# Long non-coding RNA CCRR controls cardiac conduction via regulating intercellular coupling

Yong Zhang [1], Lihua Sun[1], Lina Xuan[1], Zhenwei Pan[1], Xueling Hu[1], Hongyu Liu[2], Yunlong Bai[1], Lei Jiao[1], Zhange Li[1], Lina Cui[1], Xiaoxue Wang[1], Siqi Wang[1], Tingting Yu[1], Bingbing Feng[1], Ying Guo[1], Zonghong Liu[2], Weixin Meng[2], Hequn Ren[1], Jiyuan Zhu[3], Xuyun Zhao[1], Chao Yang[1], Ying Zhang[1], Chaoqian Xu[4], Zhiguo Wang[1], Yanjie Lu[1], Hongli Shan[1] & Baofeng Yang [1,5]

Long non-coding RNAs (lncRNAs) have emerged as a new class of gene expression regulators playing key roles in many biological and pathophysiological processes. Here, we identify cardiac conduction regulatory RNA (CCRR) as an antiarrhythmic lncRNA. CCRR is downregulated in a mouse model of heart failure (HF) and in patients with HF, and this downregulation slows cardiac conduction and enhances arrhythmogenicity. Moreover, CCRR silencing induces arrhythmias in healthy mice. CCRR overexpression eliminates these detrimental alterations. HF or CCRR knockdown causes destruction of intercalated discs and gap junctions to slow longitudinal cardiac conduction. CCRR overexpression improves cardiac conduction by blocking endocytic trafficking of connexin43 (Cx43) to prevent its degradation via binding to Cx43-interacting protein CIP85, whereas CCRR silence does the opposite. We identified the functional domain of CCRR, which can reproduce the functional roles and pertinent molecular events of full-length CCRR. Our study suggests CCRR replacement a potential therapeutic approach for pathological arrhythmias.

[1] Department of Pharmacology, State-Province Key Laboratories of Biomedicine-Pharmaceutics of China, Key Laboratory of Cardiovascular Medicine Research, Ministry of Education, College of Pharmacy, Harbin Medical University, Harbin, Heilongjiang 150081, P.R. China. [2] Department of cardiovascular surgery, The First Affiliated Hospital, Harbin Medical University, Harbin, Heilongjiang 150081, P.R. China. [3] Department of pathology, The First Affiliated Hospital, Harbin Medical University, Harbin, Heilongjiang 150081, P.R. China. [4] Mu Danjiang Medical University Mu Danjiang, Heilongjiang 157011, P.R. China. [5] Department of Pharmacology and Therapeutics, Melbourne School of Biomedical Sciences, Faculty of Medicine, Dentistry and HealthSciences University of Melbourne, Melbourne VIC 3010, Australia. These authors contributed equally: Yong Zhang, Lihua Sun, Lina Xuan. Correspondence and requests for materials should be addressed to H.S. (email: shanhongli@ems.hrbmu.edu.cn) or to B.Y. (email: yangbf@ems.hrbmu.edu.cn)

In order for heart to maintain efficient ejection of blood, cardiac muscles (cardiomyocytes) operate as a single functional syncytium, that is, they contract in unison[1]. This is because individual cardiac muscle cells are connected by the highly organized intercalated discs where gap junctions are concentrated allowing for well-orchestrated spatial propagation of action potentials through tight cardiomyocyte–cardiomyocyte coupling. Intercellular conduction of excitation through gap junctions, or more specifically gap junction channel proteins, forms electrical coupling that is essential for the functional myocardial syncytium thereby the normal impulse propagation throughout the heart[1,2]. Impaired electrical coupling promotes arrhythmogenicity and consequently contractile dysfunction. In myocardium, gap junctions exist as plaques of hexameric arrays containing hundreds to thousands of connexins, particularly connexin43 (Cx43), the pore forming subunits of gap junction channels that create a conduit connecting the cytoplasm between cells[3–5]. A hallmark of the electrical disorders with regard to impulse conduction in heart failure (HF) is the electrical uncoupling due to diminished presence of Cx43 in gap junctions, resulting in slowed conduction velocity (CV) and dispersed impulse propagation leading to an increased risk of re-entrant excitation, predisposing to cardiac arrhythmia and even sudden cardiac death[3–5].

Recently, long noncoding RNAs (lncRNAs) have emerged as a new player of gene expression regulation and coordination[6–8]. These newly recognized regulatory molecules are mRNA-like transcripts ranging from 200 nt to 100 kb in length, featured by lack of protein-coding activity, yet can participate in many fundamental biological processes and pathophysiological events. In particular, the demonstration of their roles in human cancer development and neuronal disease has garnered tremendous research interest in their possible roles in cardiovascular disease[9–11]. While a number of lncRNAs have been documented to participate in the development of HF[12–16], the possible role of these noncoding RNAs in shaping cardiac electrophysiology has not been thus far revealed.

In our pilot studies, we analyzed the microarray profiling of lncRNAs expression in a mouse model of pressure-overload HF characterized by electrophysiological disturbance with slowed cardiac conduction and increased propensity of ventricular arrhythmias. A multitude of lncRNAs and mRNAs are differentially expressed to significant levels[16]. Bioinformatics analyses (GO analysis, pathway analysis, and lncRNAs-mRNAs co-expression network analysis) enabled us to identify a subset of lncRNAs of high profiles linking to HF[16]. Among these, AK045950 stands out as a unique lncRNA that has the potential to impose impacts on the development of HF, and the associated electrical remodeling process. We, therefore, conducted the present study to explore the role of this lncRNA in regulating cardiac conduction and the associated arrhythmogenesis and to decipher the underlying cellular and molecular mechanisms. Since AK045950 is the first lncRNA identified to have the ability to control cardiac conduction, we named it CCRR. We report here that CCRR is downregulated in a mouse model of HF and in patients with HF, and this downregulation slows cardiac conduction and enhances arrhythmogenicity in HF. Moreover, CCRR silencing induces arrhythmias in healthy mice. CCRR overexpression eliminates these detrimental alterations. HF or CCRR knockdown causes destruction of intercalated discs and gap junctions to slow longitudinal cardiac conduction. CCRR overexpression improves cardiac conduction by blocking endocytic trafficking of Cx43 to prevent its degradation via binding to Cx43-interacting protein CIP85, whereas CCRR silence does the opposite. We also identified the functional domain of CCRR, which reproduces the functional roles and molecular events of the full-length CCRR. We conclude that CCRR is an antiarrhythmic lncRNA that acts by maintaining functional myocardial syncytium and its downregulation serves as a mechanism for the electrical disturbances and contractile dysfunction typically found in HF. Hence, CCRR replacement might be considered a prospective therapeutic strategy for arrhythmias under pathological conditions of the heart.

## Results

**Deregulation of CCRR slows cardiac conduction.** Expression of CCRR was found robustly downregulated (~55%) in the myocardium of HF mice (Fig. 1a). Notably, the human homolog to CCRR was also enormously reduced in RNA samples from individuals with HF compared with healthy subjects (Fig. 1b; see Supplementary Table 1 for patient information, and Supplementary Figure 1 for the conserved sequence motif between mouse and human). The results generated by fluorescence in situ hybridization (FISH) experiments (Fig. 1c) and by Northern blot analysis (Fig. 1d) confirmed the quantitative polymerase chain reaction (qPCR) data, uncovering ~75% decrease in CCRR level in HF.

To investigate whether downregulation of CCRR has protective or detrimental consequences, we assessed the effects of CCRR on cardiac conduction and the associated arrhythmogenicity and cardiac contractile function in HF mice with both gain- and loss-of-function approaches. The results acquired using the optical mapping with voltage-sensitive dyes showed that the CV between the apex and the base for longitudinal propagation was significantly slowed in HF hearts (Fig. 2a). Similar results showing deceleration of conduction in HF was also obtained by analysis of surface ECG recordings in Langendorff-perfused hearts (Supplementary Figure 2a, b). Consistently, the generation of arrhythmias in response to programmed left ventricular tachypacing was significantly facilitated, as indicated by the increase in the incidence of ventricular tachycardia (VT) and prolongation of VT duration in HF mice (Fig. 2b). Meanwhile, cardiac contractile function was also impaired, as reflected by the decreased ejection fraction (EF), and fractional shortening (FS) (Supplementary Figure 3), as well as the decreased left ventricular wall thickness, indicating the state of HF (Supplementary Table 2 for echocardiographic characterization of HF phenotypes). Strikingly, all these deleterious alterations were significantly mitigated in HF mice pretreated with the lentivirus carrying the CCRR gene (Lv-CCRR) for overexpression (Fig. 2a, b; Supplementary Figure 3a; Supplementary Table 3). Intracavity administration of Lv-CCRR nearly abolished the electrical disturbances: acceleration of cardiac conduction and suppression of VT. The negative control lentivirus (Lv-NC) failed to elicit these beneficial effects. The efficiency of lentivirus infection was confirmed by the robust overexpression of CCRR in the myocardium (Supplementary Figures 4 and 5).

We reasoned that if downregulation of CCRR indeed contributes to the observed pathophysiological processes in our model, then artificial knockdown of this lncRNA in otherwise healthy animals should be able to reproduce the electrophysiological and contractile phenotypes of HF. Our experiments on mice that received the lentivirus containing CCRR-specific siRNA (Lv-siCCRR) via heart cavity injection after clipping the aorta indeed generated the evidence in support of this thought (Supplementary Table 4). As illustrated in Fig. 2c, d, CCRR knockdown by Lv-siCCRR decelerated cardiac conduction (also see Supplementary Figure 2c), and increased the likelihood of arrhythmia induction in normal mice. Consequently, Lv-siCCRR impaired cardiac contractile function (Supplementary Figure 3b). The effectiveness of the siRNA to knockdown endogenous CCRR was verified under both in vivo and in vitro conditions

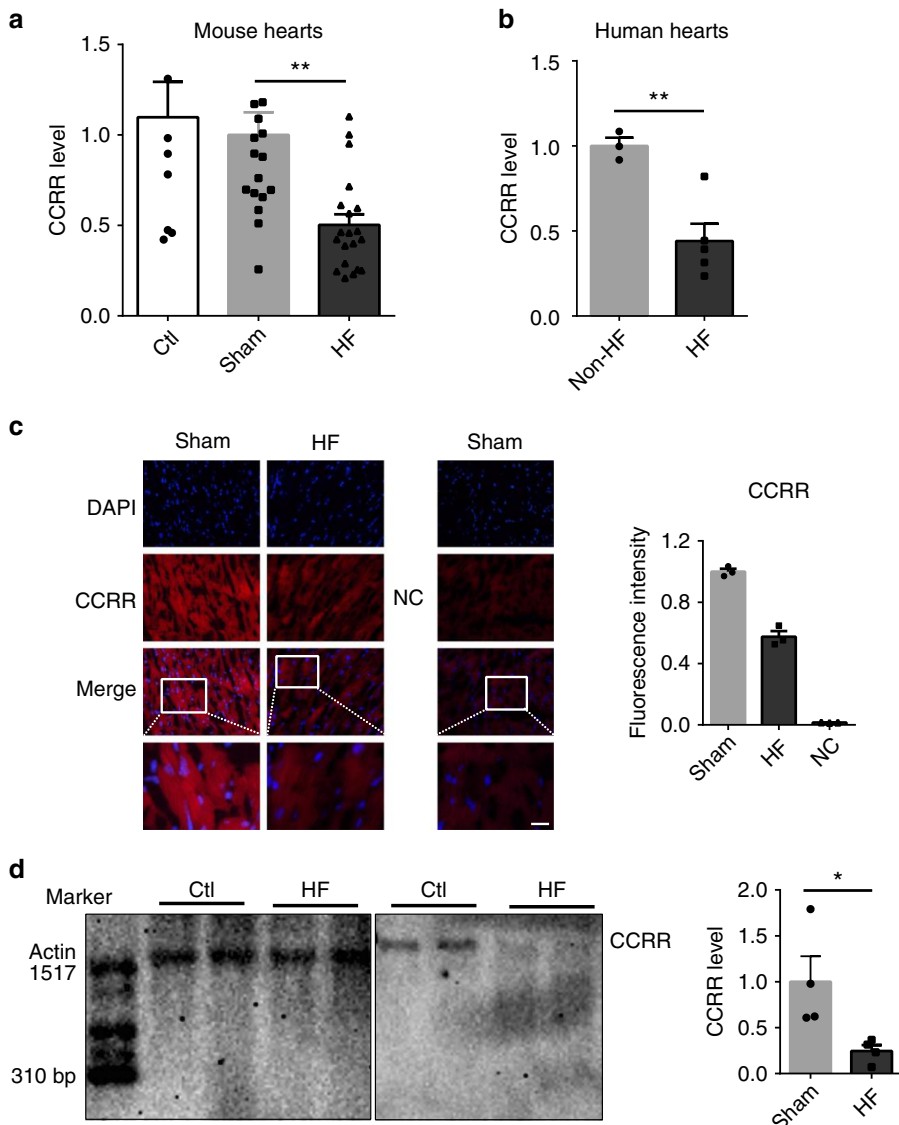

**Fig. 1** Expression deregulation of lncRNA-CCRR in heart failure (HF). **a** Real-time RT-PCR (qPCR) results showing the decrease of CCRR (cardiac conduction regulating RNA) expression in a mouse model of HF mice. $^{**}P < 0.01$ HF vs. Sham-control; $n = 18$ mice for each group. **b** Downregulation of CCRR expression in patients with HF as compared to non-HF human subjects, determined by qPCR. $^{**}P < 0.01$ HF ($n = 5$) vs. non-HF; $n = 3$ patient samples for each group. **c** Fluorescence in situ hybridization (FISH) images (200× magnification) showing the decrease of CCRR expression in HF mice. Left panel: representative images showing the reduced signal intensity (red) indicative of reduced CCRR level in the cytoplasm. Right panel: averaged values of fluorescence intensity for quantification of CCRR expression. NC: negative control probe. $^{**}P < 0.01$ HF vs. Sham-control; $n = 3$ hearts for each group. **d** Northern blot analysis showing the decrease of CCRR expression in HF mice. $^{*}P < 0.05$ HF vs. Sham; $n = 4$ mice for each group. (Mean ± SEM; analysis of variance—ANOVA followed by Dunnett's test for comparisons among multiple groups, and Student $t$ test for comparisons between two groups)

(Supplementary Figures 4 and 5). In all situations, the negative control siRNA (Lv-siNC) failed to produce any meaningful effects. These data suggest that CCRR is an antiarrhythmic lncRNA that acts by maintaining normal cardiac conduction and downregulation of CCRR is sufficient to cause conduction slowing leading to enhanced propensity of arrhythmias, which may account at least partly for the electrical disorders in HF.

**Cellular and subcellular mechanisms for the action of CCRR.** To elucidate the possible cellular and subcellular mechanisms for the antiarrhythmic properties of CCRR, we performed electron microscopic examinations on the changes of myocardial microstructure induced by HF or by Lv-CCRR or Lv-siCCRR. As

shown in Fig. 3a, HF caused significant mitochondria swelling and gap junction rupturing in mice (upper left and middle panels) and patients (right panels). As anticipated, Lv-CCRR largely mitigated the adverse alterations of mitochondria and gap junction in HF mice (Fig. 3a, lower left). These detrimental alterations are consistent with the observed slowing of cardiac conduction and also indicate the possibility of uncoupling of intercellular propagation of cardiac excitations as a mechanism for the conduction slowing and enhanced arrhythmogenesis[17–20].

The results presented above strongly suggested that the loss of gap junction integrity contributes to the deleterious changes of cardiac electrical and contractile function as a subcellular mechanism in our HF model. We then went on to investigate the changes of the protein level of Cx43 and CIP85 (Cx43-interacting protein of 85 kDa)[21–23] in the cytoplasmic membrane.

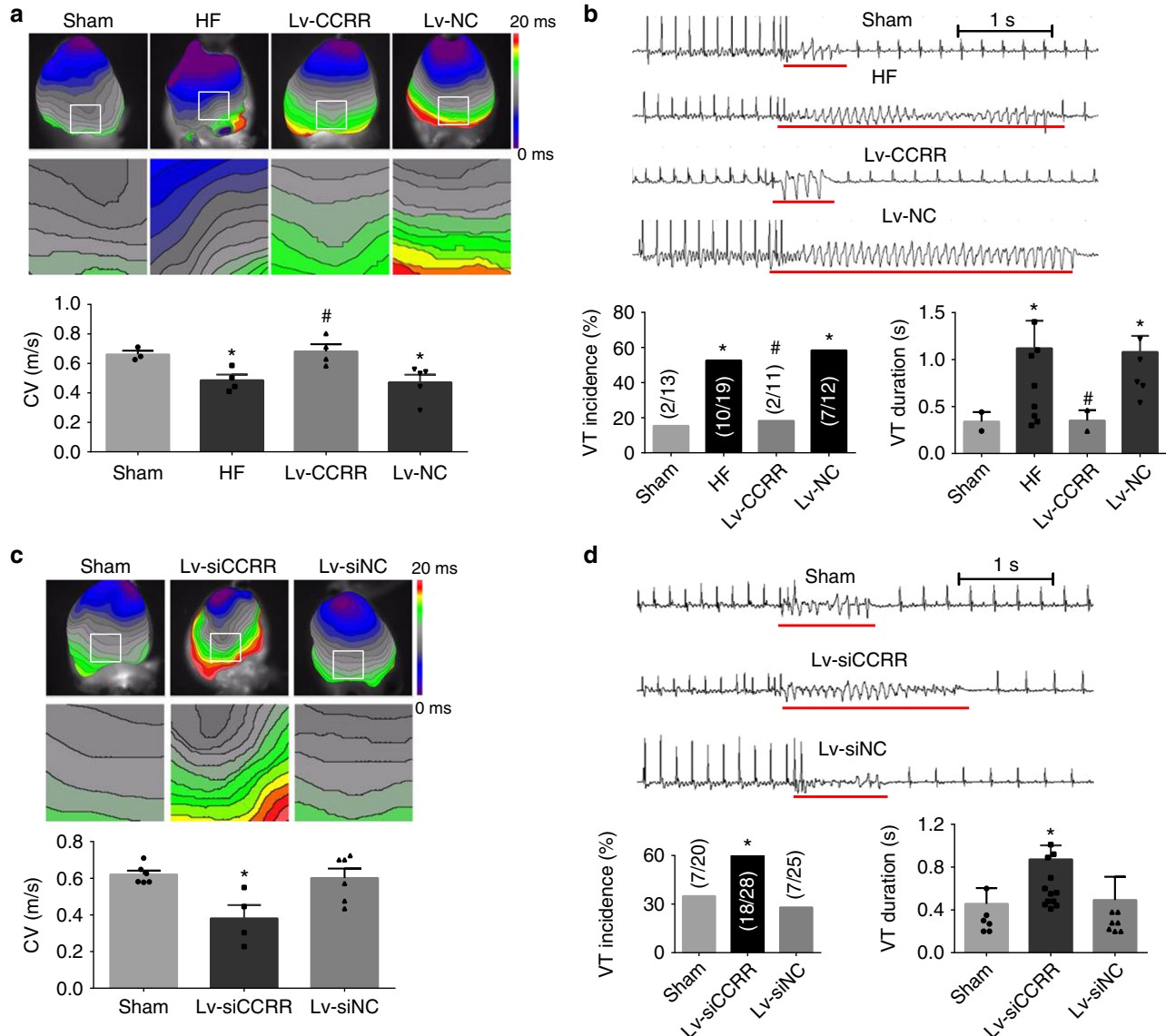

**Fig. 2** Regulation of cardiac conduction and arrhythmias by CCRR in HF mice. **a** Slowing of conduction velocity (CV) in HF and restoration by CCRR overexpression. CV was determined by optical mapping techniques with a voltage-sensitive dye to define the cardiac activation. CV values were calculated from the gradient of the scalar field of 12-ms isochronal activation maps along the septal apex–base axis: CV = distance/12 ms. Note that CV was substantially decreased in HF and this conduction slowing was restored in the hearts pretreated with the lentivirus carrying the CCRR gene for overexpression (Lv-CCRR), but not with the negative control construct (Lv-NC). Viral vectors were administered by intracavity injection (directly injected into the left ventricular chamber). The Sham group underwent the same surgical procedures without TAC. $^*P < 0.05$ HF or Lv-NC vs. Sham-control; $^\#P < 0.05$ Lv-CCRR vs. HF; $n = 4$ mice for each group. **b** Antiarrhythmic effects of CCRR overexpression in a mouse model of HF. The incidence and duration of ventricular tachycardia (VT) induced by programmed stimuli were determined from ECG recordings. Note that the arrhythmogenicity was significantly enhanced in HF hearts, which was considerably suppressed by Lv-CCRR, but not by Lv-NC. The red lines in the ECG traces indicate VT duration, and the values within the parentheses in the bar charts indicate VT incidence. $^*P < 0.05$ HF or Lv-NC *vs.* Sham-control; $^\#P < 0.05$ Lv-CCRR vs. HF. **c** Slowing of cardiac conduction induced by CCRR knockdown in healthy mice. The lentivirus vector engineered to contain a CCRR siRNA (Lv-siCCRR) was injected into the left ventricular chamber to silence myocardial CCRR. Lv-siCCRR caused a remarkable decrease in cardiac CV, whereas the negative control (Lv-siNC) failed to elicit any significant changes. The Sham group received the same surgical procedures without injecting vectors. $^*P < 0.05$ Lv-siCCRR vs. Sham-Control; $n = 4$ mice for each group. **d** Pro-arrhythmic effects of CCRR knockdown in healthy mice. A prominent finding here is that knockdown of endogenous CCRR by Lv-siCCRR was sufficient to induce VT in otherwise healthy hearts. $^*P < 0.05$ Lv-siCCRR vs. Sham-Control. (Mean ± SEM; ANOVA followed by Dunnett's test for multiple group comparisons, Student $t$ test for two group comparisons, and $\chi^2$-test for nonparametric data set comparisons)

As shown in Fig. 3b, the protein level of Cx43 was remarkably decreased in the plasma membrane but significantly increased in the cytoplasm resulted in reduction of total Cx43 protein level in HF mice relative to the Sham-Control animals (also see Supplementary Figure 6a). Similar pattern of changes of CIP85 was observed, with an exception that the total protein level of

CIP85 remained unaltered in HF (Supplementary Figure 6b). Notably, Lv-CCRR effectively corrected these HF-induced abnormal changes (Fig. 3b).

On the other hand, knockdown of endogenous CCRR by Lv-siCCRR simulated the HF-induced abnormal changes of intercalated discs and gap junctions in otherwise normal mice

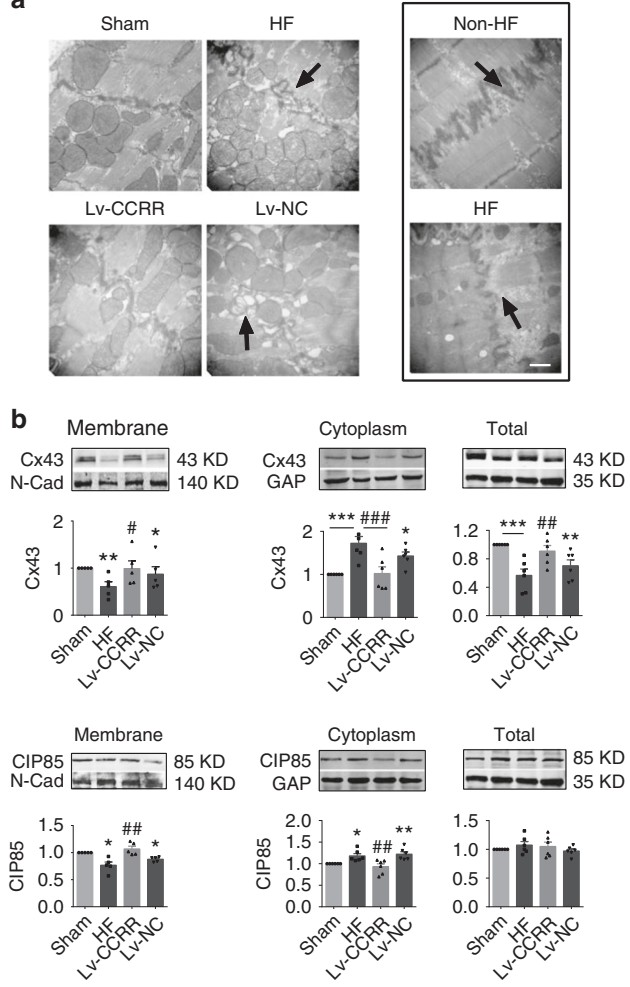

**Fig. 3** CCRR alleviates destruction of intercalated discs and gap junction in HF mice. **a** Typical examples of electron microscopic images (30,000× magnification) showing the deranged intercalated discs in HF mice (left panels) and patients (right panels), and the effects of CCRR overexpression by Lv-CCRR on the microstructure of left ventricular cardiac muscles in HF mice. Marked disruption of intercalated discs and gap junction was consistently observed in HF myocardium (indicated by arrows), and such derangement was ameliorated and restored to normal conditions after pretreatment with Lv-CCRR, but not with Lv-NC. Similar observations were repeated in another three HF mice. **b** Effects of CCRR overexpression by Lv-CCRR on the protein levels of connexin43 (Cx43) and CIP85 (the Cx43-interacting protein that regulates the endocytic trafficking of Cx43 for degradation) in HF mice. Upper panel: examples of Western blot bands; lower panel: averaged data on Cx43 and CIP85 protein levels. Comparisons of Cx43 and CIP85 at the protein levels were made between the membrane and cytoplasm fractions. Cx43 level was markedly decreased in the membrane, but increased in the cytoplasm, which resulted in an overall reduction of total protein level, indicating the enhanced endocytic trafficking of Cx43 in HF mice compared to the Sham-operated control animals. The membrane bands were normalized to N-cadherin (N-Cad), and the cytosolic and total protein bands were normalized to GAPDH (GAP). $^*P < 0.05$, $^{**}P < 0.01$, $^{***}P < 0.001$; $^{\#}P < 0.05$, $^{\#\#}P < 0.01$, $^{\#\#\#}P < 0.001$ Lv-CCRR vs. HF; $n = 5$ heart samples of membrane fraction for each group, $n = 6$ for cytoplasmic fraction, and $n = 6$ for total protein. Similar changes were observed with CIP85, but its total protein level remained unaffected. $^*P < 0.05$, $^{**}P < 0.01$; $^{\#\#}P < 0.01$, Lv-CCRR vs. HF; $n = 5$ heart samples of membrane fraction for each group, $n = 6$ for cytoplasmic fraction, and $n = 6$ for total protein

(Fig. 4a), which was not seen with Lv-siNC. Likewise, the normal mice pretreated with Lv-siCCRR demonstrated considerable reductions of Cx43 and CIP85 levels in the cytoplasmic membrane and increases in the cytoplasm as those seen with HF hearts (Fig. 4b). The same pattern of changes of CIP85 and Cx43 indicates the possible physical association between the two proteins.

Our immunohistochemistry results showed the expected localization of Cx43 to the cell periphery in the form of punctuates reminiscent of gap junction plaques in tissue, and CIP85 was also localized to the gap junction membrane with some intracellular punctuate reactions (Fig. 5a for mouse and Fig. 5b for human). Merged images revealed the co-localization of Cx43 with CIP85 in regions of gap junction plaques. Immuno-histochemical staining consistently showed reduced membrane localization of Cx43 as indicted by the diminished density of fluorescence signals at intercalated discs of HF hearts of both mice (Fig. 5a) and humans (Fig. 5b) or in the mouse hearts pretreated with Lv-siCCRR, and conversely increased expression of Cx43 when pretreated with Lv-CCRR (Fig. 5a).

The effects of CCRR on CIP85 and Cx43 were reproduced in cultured neonatal mouse ventricular myocytes (NMVMs). As shown in Supplementary Figure 7, CCRR overexpression increased the presence of Cx43 in the cytoplasmic membrane, whereas CCRR knockdown decreased it. The relationship between CIP85 and Cx43 was also verified by Western blot and immunocytostaining analyses (Supplementary Figure 8). Trans-fection of CIP85 siRNA (siCIP85) elevated the protein level of Cx43 while silencing CIP85.

We then further confirmed the role of CCRR in regulating intercellular communications (presumably through gap junction channels) using a diffusible dye (Lucifer yellow) in adult human ventricular cardiomyocyte AC16 cells (Fig. 6). Lv-CCRR significantly increased, whereas Lv-siCCRR nearly completely blocked, the diffusion of Lucifer yellow among AC16 cells. The results provided another line of evidence for the critical role of CCRR in maintaining overall cardiac conduction.

Cardiac conduction is determined by both intracellular conduction and intercellular propagation of excitation waves. While Cx43 is responsible for the latter, sodium channel determines intracellular CV. We thus also assessed the effects of CCRR on Nav1.5, the primary α-subunit of cardiac sodium channels in HF mice. Our results demonstrated that Nav1.5 protein level was upregulated in the healthy mice pretreated with Lv-siCCRR to knockdown endogenous CCRR, but downregulated when treated with Lv-CCRR for CCRR overexpression (Supplementary Figure 9). These results indicated possible acceleration of intracellular conduction, presumably as a compensatory mechanism for the loss of intercellular conduction and excluded Nav1.5 as a mechanistic link to the observed conduction disturbance in our model.

**Molecular mechanisms for the regulation of Cx43 by CCRR.** We then took a further step to decipher how CCRR regulates Cx43 expression in the membrane. Our data from fluorescence in situ hybridization indicated that the cytoplasm is the primary workspace for CCRR as it is primarily localized in the cytoplasm (Fig. 1c). It has been documented that CIP85 regulates the endocytic trafficking of Cx43 from the plasma membrane to induce its degradation by the lysosome-mediated mechanism[21–23]. Thus, there is a possibility for CCRR to bind CIP85 to interfere the CIP85:Cx43 interactions. This is indeed supported by our theoretical analysis of RNA:protein binding using the RNA–protein interaction prediction (RPIseq) database,

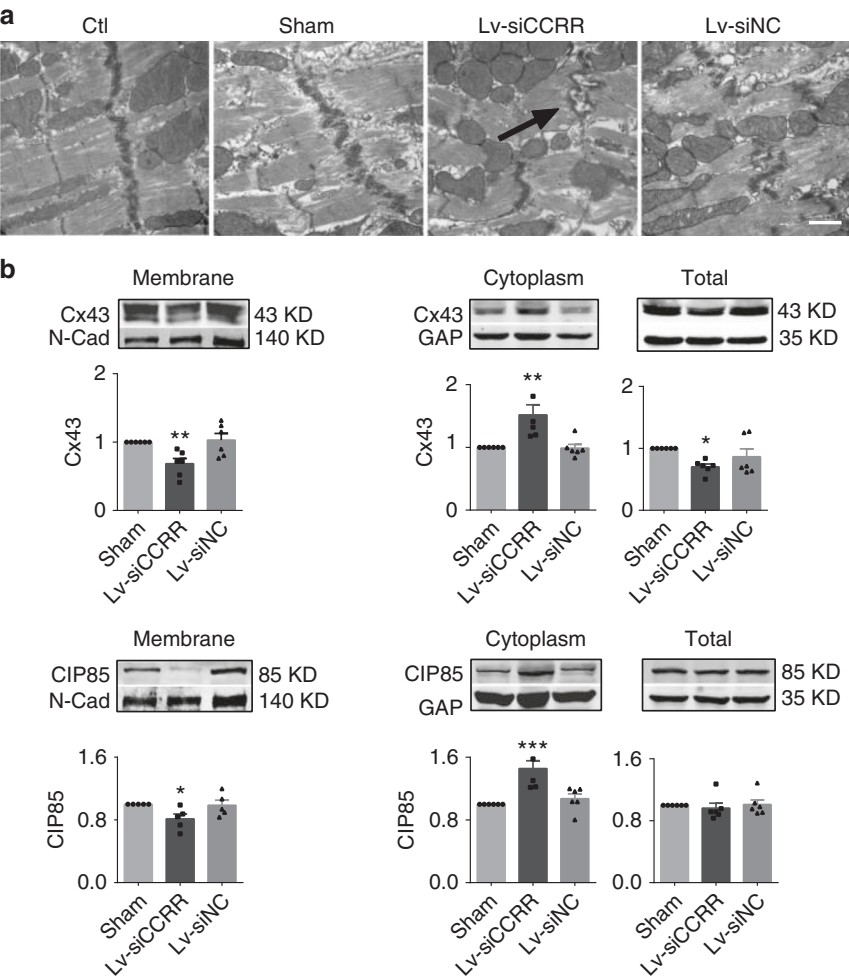

**Fig. 4** CCRR knockdown causes destruction of intercalated discs and gap junction in healthy mice. **a** Representative images of electron microscopy (30,000× magnification) showing the effects of knockdown of endogenous CCRR by Lv-siCCRR on the microstructure of left ventricular cardiac muscles in healthy mice. Note the profound derangement and loss of intercalated discs in the hearts pretreated with Lv-siCCRR (dark arrows), resembling those noted in HF heart, which was not observed with Lv-siNC and Sham-Control (subjected to the surgical procedures but without lentivirus injection). Similar results were observed in another three independent experiments. **b** Effects of CCRR knockdown by Lv-siCCRR on the protein levels of Cx43 (left panel) and CIP85 (right panel) in healthy mice, with comparisons between the membrane and cytoplasm fractions. Lv-siCCRR elicited similar alterations of Cx43 and CIP85 levels in normal hearts as those seen above with HF hearts. Cx43 was decreased in the membrane fraction, but increased in the cytoplasm, with a net reduction of total protein levels. $^*P < 0.05$, $^{**}P < 0.01$, $^{***}P < 0.001$ Lv-siCCRR vs. Sham; $n = 6$ mice for each group. CIP85 showed similar changes but with the total protein level unaffected. $^*P < 0.05$, $^{**}P < 0.01$ Lv-siCCRR vs. Sham; $n = 5$ heart samples of membrane fraction for each group, $n = 6$ for cytoplasmic fraction, and $n = 6$ for total protein. (Mean ± SEM; ANOVA followed by Dunnett's test for multiple group comparisons, and Student $t$ test for two group comparisons)

which revealed a high probability of CCRR:CIP85 interaction (Supplementary Figure 10). To experimentally verify this issue, we first carried out RNA-binding protein immunoprecipitation (RIP) analysis to see if CCRR could bind CIP85. The results depicted in Fig. 7a clearly indicate the presence of such an interaction: immunoprecipitation (IP) of CIP85 specifically retrieved a robust amount of CCRR, and conversely RNA pull-down of CCRR dragged down an impressive quantity of CIP85, with an unrelated lncRNA *ZFAS1* showing the negative result. We next assessed the alterations of the endogenous CIP85:Cx43 interactions by co-immunoprecipitation (Co-IP). As exhibited in Fig. 7b, the anti-CIP85 antibody identified significant CIP85 protein in the Cx43-immunoprecipitated samples, and vice versa Cx43 was picked up by its antibody from the CIP85-immunoprecipitated samples. Intrudingly, the predicted binding motif is within the well-conserved sequence domain (Supplementary Figures 1 and 10c), suggesting the functional role of this sequence domain. To test this point and get insight into the

molecular mechanisms for the actions of CCRR, we investigated the functional role of a CCRR fragment (290 nts) encompassing the predicted binding site (Supplementary Figure 10c) by the following steps. First, we verified the ability of this fragment (for convenience, we named it FD for functional domain) to bind CIP85 using RIP and pulldown methods. As depicted in Fig. 7c, CIP85 antibody dragged down significant amount of FD. On the other hand, the CIP85 protein pulled down by FD after infection of mouse heart with a lentiviral vector carrying FD (Lv-FD) was significantly increased relative to that with the empty viral vector or with a lentiviral vector carrying a negative control sequence (Fig. 7d).

While the above results indicate that FD is sufficient to bind CIP85, we wanted to know if this fragment could reproduce the effects of the full-length CCRR on cardiac conduction. To this end, we looked the changes of cardiac CV and arrhythmias in a mouse model of acute myocardial infarction (MI) after infection with Lv-FD. The results consistently showed that FD, similar to

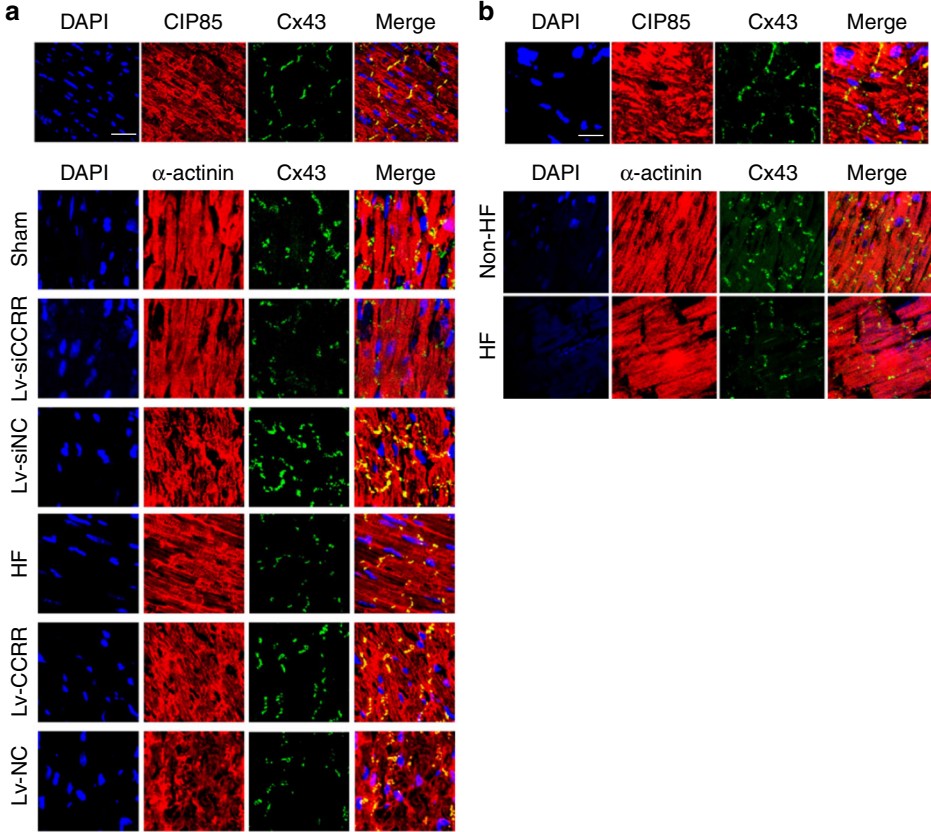

**Fig. 5** Effects of CCRR on the gap junction distribution of Cx43 and CIP85. **a** Immunostaining of cardiac slices showing the co-localization (yellow) of CIP85 (red) and Cx43 (green) proteins at the intercalated discs in healthy hearts (upper panel), and the effects of CCRR on the gap junction distribution of these proteins (600× magnification, lower panel). Knockdown of endogenous CCRR by Lv-siCCRR drastically diminished the presence of Cx43 (stained in green) in the intercalated discs in healthy mouse hearts, resembling the changes caused by HF. In contrast, CCRR overexpression by Lv-CCRR normalized the HF-induced expression repression of Cx43. DAPI (blue) was used to stain nuclei and α-actinin (red) to identify the cell contour. Similar results were observed in another three separate experiments. **b** Immunostaining of cardiac slices showing the co-localization of CIP85 (stained in red) and Cx43 (green) proteins at the intercalated discs in healthy control subjects (600× magnification). Cx43 in the intercalated discs in HF patient significantly diminished compared with healthy control subjects. DAPI (blue) was used to stain nuclei and α-actinin (red) to identify the cell contour. Similar results were observed in another two separate experiments

CCRR, rescued the delay of cardiac conduction induced by MI and suppressed the ischemic arrhythmias (Supplementary Figure 11).

Moreover, electron microscopic examination displayed that FD relieved the destruction of gap junctions induced by MI (Fig. 8a). Furthermore, while MI caused a significant decrease in Cx43 total protein level, FD recovered such a deleterious change (Fig. 8b). As a comparison, the negative construct did not elicit any appreciable beneficial effects.

As a more straightforward evidence for the role of FD on cardiac conduction, transfection of a plasmid containing FD (pFD) significantly increased the diffusion of Lucifer yellow between adult human ventricular cardiomyocyte AC16 cells (Fig. 9a), consistent with the effect of the full-length CCRR (Fig. 6). In all cases, the mutation of the nucleotides responsible for binding with CIP85 in FD (pFD-Mut; Supplementary Figure 10c) rendered a loss-of-function in regulating intercellular communications, further indicating the key role of binding of FD or CCRR to CIP85 to improve gap junction in maintaining and accelerating cardiac conduction.

To verify that changes of CCRR or FD level could indeed interfere the interaction between Cx43 and CIP85, we looked at the effects of FD overexpression on the Cx43 level pulled down by CIP85 using Co-IP analysis. As illustrated in Fig. 9b,

in cultured NMVMs transfected with the pFD for over-expression, the protein level of Cx43 co-immunoprecipitated by CIP85 antibody was significantly reduced. These data provided a line of evidence that the FD of CCRR binds to CIP85 and subsequently weakens the interaction between Cx43 and CIP85.

Since CIP85 is known to induce the endocytic trafficking of Cx43 for degradation by the lysosome-mediated mechanism[21–23], we decided to investigate if functional inhibition of CIP85 binding by FD of CCRR could reduce Cx43 degradation in lysosomes. For this purpose, we conducted immunocytostaining of Cx43 and Lamp1 (the biomarker for lysosome). As illustrated in Fig. 9c, Cx43 was found localized primarily to the plasma membrane and lysosomes in the cytoplasm, and under control conditions there was only a small amount of Cx43 in the lysosomes as indicated the low level of overlapping staining between Cx43 and Lamp1. Strikingly, either pCCRR or pFD wiped out the overlapping staining indicating the decreased recruitment of Cx43 into lysosomes for degradation, and simultaneously enhanced the Cx43 staining to the cytoplasmic membrane indicating the increased presence of Cx43 in the plasma membrane. In sharp contrast, siCCRR significantly increased the level of over-lapping staining between Cx43 and Lamp1, whereas pCCRR-

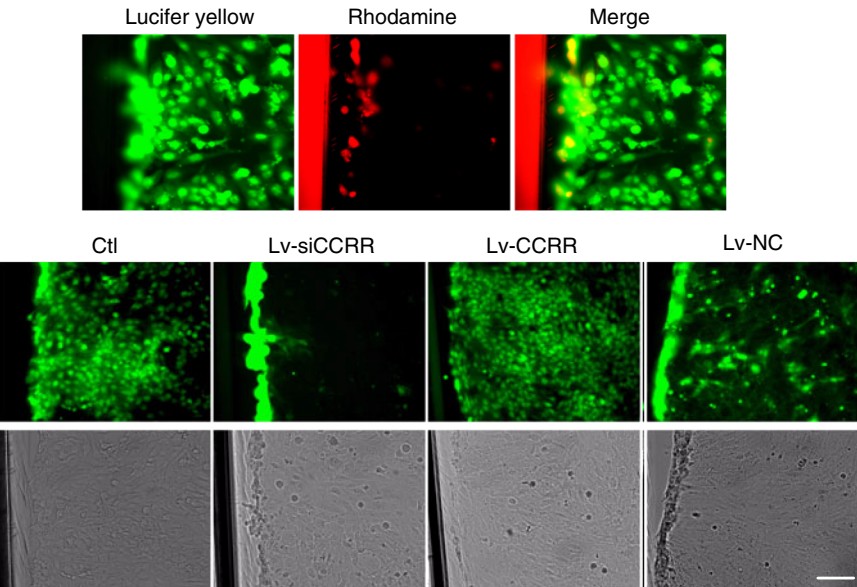

**Fig. 6** CCRR improves gap junction communication in AC16 human heart cells. AC16 human adult ventricular cardiomyocytes were loaded with a diffusible fluorescent dye (Lucifer yellow), and the diffusion of the dye was monitored under a laser confocal microscope (200× magnification). Note that CCRR overexpression after infection of Lv-CCRR significantly promoted, whereas CCRR knockdown by Lv-siCCRR nearly completely blocked, the diffusion of Lucifer yellow among AC16 cells (green). The high molecular weight marker dye conjugate Rhodamine B was used as a negative control (red). Similar data were acquired from another three independent experiments

NC and pFD-Mut did not caused any significant alterations relative to control.

## Discussion

Collectively, we made a number of novel findings in the present study. (1) CCRR is abnormally downregulated in a mouse model of HF, which was accompanied by cardiac conduction slowing and contractile dysfunction. (2) Downregulation of CCRR alone is sufficient to induce electrophysiological disturbances and syncytium disruption leading to contractile dysfunction by creating intercellular uncoupling through disrupting gap junction. This is supported by the results that artificial knockdown of CCRR reproduced conduction slowing and the associated arrhythmogenesis in healthy mice, which could be reversed by CCRR replacement. (3) On the contrary, overexpression of CCRR abrogates the conduction anomalies in the setting of HF, and CCRR was required for maintaining proper distribution of Cx43 in the intercalated discs by binding to CIP85 protein so as to prevent Cx43 from backward trafficking and subsequent degrading, which likely served as a downstream mechanism for the disruption of gap junction (Fig. 10). (4) A sequence domain of CCRR that is conserved across species is responsible for CCRR: CIP85 interaction and the consequent disruption of CIP85:Cx43 interaction is sufficient to produce the beneficial actions as the full-length CCRR did on cardiac conduction and the associated arrhythmias. On the basis of these findings, we conclude that CCRR is an antiarrhythmic lncRNA that acts by maintaining functional myocardial syncytium and its downregulation serves as a mechanism for the electrical disturbances and contractile dysfunction typically found in HF.

Probably the most notable finding was that downregulation of CCRR resulted in severe disruption of gap junction and electrical uncoupling leading to impairment of myocardial syncytium. One of the consequences of such adverse changes was the enhanced propensity of ventricular arrhythmias due to discontinuation of intercellular excitation propagation and slowing of overall cardiac conduction. Another aspect of the deleteriousness brought about by slowed cardiac conduction was weakened contractile function due to the loss of functional myocardial syncytium[24–27]. Our study represents the first to identify a lncRNA with the antiarrhythmic property. These findings strongly indicate the critical involvement of lncRNAs in controlling cardiac electrophysiology, which should tremendously advance our understanding of the cellular functions and pathophysiological roles of lncRNAs. Yet, it should be mentioned that CCRR might also be able to act directly on the structural proteins of cardiomyocytes or other cells that are important to heart function and intercalated disc maintenance. Our study does not in any way exclude such a possibility that merits future studies to clarify.

Identification of CCRR loss-of-function as a new mechanism for the development of HF not only aids us to unravel the pathophysiological role of lncRNAs in cardiovascular disease, but also indicates a new approach that we could employ to handle the provoked arrhythmogenesis in the setting of HF. Our findings of the beneficial effects afforded by CCRR gain-of-function strongly suggest that CCRR replacement is a prospective therapeutic strategy for HF, particularly when considering the similar downregulation of CCRR and Cx43 and disruption of gap junction in the hearts of HF patients unraveled in this study. Our results indicated that a 290-nt fragment of CCRR was able to reproduce the beneficial actions of CCRR lends a further support to this view. Indeed, Cx43 dysfunction is a common mechanism for electrical disorders encountered in a variety of cardiovascular diseases including myocardial ischemia, cardiomyopathies, cardiac hypertrophy, atrial fibrillation, etc.[3–5], in addition to HF, and improving cardiac gap junction communication has been considered a new antiarrhythmic mechanism[2,18–20]. Nonetheless, extensive future studies are required to validate the potential of CCRR.

While lncRNAs are known to be associated with >150 human diseases, based on the annotations from the lncRNA and disease databases[28], their role in cardiovascular disease has just begun to be uncovered and the pertinent experimental studies have been rather scant. Here, we have presented an in-depth experimental

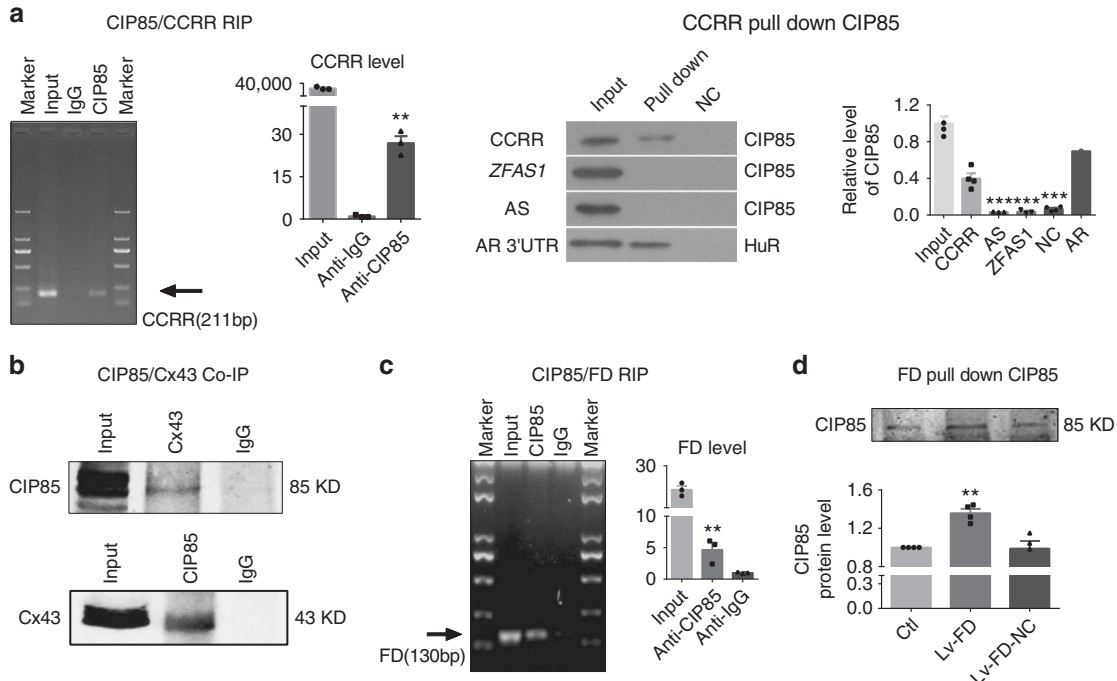

**Fig. 7** Interactions between CCRR and CIP85 and between CIP85 and CX43. **a** RNA immunoprecipitation (RIP) showing the physical interaction between CCRR and CIP85 proteins in cultured neonatal mouse ventricular myocytes (NMVMs). Most left panel: the electrophoresis gel image showing the anticipated fragment representing CCRR that was pulled down by CIP85 antibody with IgG antibody as a negative control. Middle left panel: the quantified results of the electrophoretic bands. $^{**}P < 0.01$ anti-CIP85 vs. anti-IgG; $n = 4$ batches of cells for each group. Middle right panel: an example of immunoblotting bands showing the pulldown of CIP85 protein by CCRR or other RNAs as labeled on the left. ZFAS1: an lncRNA used as a negative control; AS: antisense to CCRR used as another negative control; AR: androgen receptor 3' UTR: a commercially obtained positive control; Input: purified CIP85 as a positive control; NC: negative control probe; HuR: Hu-antigen R. Right panel: mean relative band densities representing CIP85 protein with varying treatments. $^{***}P < 0.001$ AS, ZFAS1, NC vs. CCRR; $n = 4$. **b** Protein co-immunoprecipitation (Co-IP) confirming the physical interaction between CIP85 and Cx43 in cultured NMVMs. Upper panel shows the presence of CIP85 in an anti-Cx43 pulldown sample, and lower panel shows the presence of Cx43 in an anti-CIP85 pulldown sample. **c** RIP results showing the physical interaction between FD and CIP85 proteins in heathy mice. Right panel: the quantified results of the electrophoretic bands. $^{**}P < 0.01$ anti-CIP85 vs. anti-IgG; $n = 3$ mice for each group. **d** Immunoblotting image showing the pulldown of CIP85 protein by FD. Sham: non-treated sample; Lv-FD: lentiviral vector carrying FD. Lv-FD-NC: lentiviral vector carrying negative sequence. Lower panel: mean relative band densities representing CIP85 protein with varying treatments. $^{**}P < 0.01$ Lv-FD vs. Lv-FD-NC; $n = 4$. (Mean ± SEM; ANOVA followed by Dunnett's test for multiple group comparisons, and Student $t$ test for two group comparisons)

study on a specific lncRNA-CCRR in the setting of HF. This study characterized the role of CCRR downregulation in inducing arrhythmic phenotypes, defined intercellular uncoupling as a subcellular mechanism for the arrhythmogenesis, and deciphered backward trafficking as a signaling mechanism for reduced Cx43 protein in the plasma membrane in HF. These findings allowed us to draw up the following signaling pathway as a new mechanism for the long-recognized yet poorly understood arrhythmogenesis in HF:HF → CCRR↓ → CIP85↑ → Cx43↓ → Gap junction↓ → Cardiac conduction↓ → Functional syncytium↓ → Arrhythmias↑ (Fig. 10).

## Methods

**Human preparations**. We obtained the human cardiac tissues from the First Affiliated Hospital of the Harbin Medical University under the procedures approved by the Ethnic Committee for Use of Human Samples of the Harbin Medical University (HMUIRB20160021), all human studies were performed in compliance with the relevant ethical regulations, The written informed consents were obtained from all subjects recruited to our study. The informed consents were obtained from every human subjects included in our study. The human ventricular endocardial tissue samples were collected from five HF patients diagnosed according to the NYHA (New York Heart Association) classification for the diagnosis and who had circulating NT-proBNP > 300 ng L$^{-1}$ or BNP > 100 ngL$^{-1}$ (Supplementary Table 1 for the clinical characteristics of the patients)[29]. Tissues from three healthy human subjects were used as a control for HF samples. These subjects died of traffic accident and had voluntarily consented on the whole-body donation for medical research before accident.

**Mouse model of pressure overload-induced congestive HF**. C57BL/6 male mice weighing 22–26 g were obtained from the Liaoning Changsheng Biotechnology Co. Ltd. All procedures involving animals and their care were approved by the Institutional Animal Care and Use Committee of Harbin Medical University (No. HMUIRB-2008-06), all animal experiments were performed in compliance with the relevant ethical regulations. Animals were randomly divided into Sham-Control, HF model groups, and other groups as specified in the result descriptions. The pressure-overload HF was created by transverse aortic constriction (TAC) following the procedures described in our previous study[16]. In brief, an incision was made at the level of the suprasternal notch to allow direct access to transverse aortric. TAC was then performed by ligating the aorta between the right innominate artery and the left carotid artery over a 26-gauge blunt needle using 5-0 silk sutures for approximately 75% aortic constriction.

Eight weeks later, the animals received echocardiographic examinations. The mice with EF% value lower than 50% cutoff were defined as HF. After 8 weeks, the mice were subjected to experimental interventions and measurements as specified in the data presentation. The animals were then sacrificed after anesthetized with 2,2,2-tribromoethanol (200 mg kg$^{-1}$, i.p.; Sigma, St. Louis, MO, USA), and their hearts were quickly excised and weighted in cold (4 °C) buffer. The left ventricle (LV) was then rapidly frozen in liquid nitrogen and stored at −80 °C for molecular experiments. In addition to the depressed cardiac contractile function, heart weight/body weight ratio and lung weight/body weight ratio were also measured to indicate the development of HF in our TAC model (Supplementary Figure 12). The investigator was blinded to the group allocation during the whole experimental procedures.

**Mouse model of acute MI**. C57BL/6 male mice weighing 22–26 g were anesthetized by intraperitoneally injected with 2% afloxide (100 mg kg$^{-1}$, ip; T48402; Sigma-Aldrich Corporation, St. Louis, MO, USA), and their chests were opened to expose the hearts. The left descending coronary artery (LAD) was ligated with a 7/0

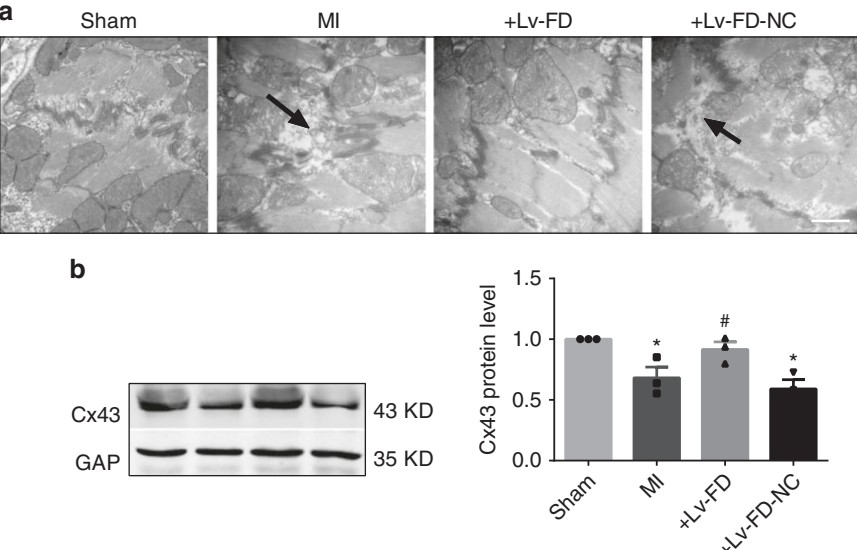

**Fig. 8** Effect of FD on gap junctional integrity. **a** Typical examples of electron microscopic images (30,000× magnification) showing the deranged intercalated discs in MI mice, and the effects of FD overexpression by Lv-FD on the microstructure of left ventricular cardiac muscles. Marked disruption of intercalated discs and gap junction was consistently observed in infarcted myocardium (dark arrows), and such derangement was ameliorated and restored to normal conditions after pretreatment with Lv-FD, but not with Lv-FD-NC. Similar observations were repeated in another three MI mice. **b** Effect of FD overexpression on the protein level of connexin43 (Cx43) in MI mice. Left panel: an example of Western blot bands; right panel: averaged data on Cx43 protein level. Cx43 expression was significantly repressed by MI and rescued by Lv-FD, relative to control and Lv-FD-NC. *$P < 0.05$ MI or Lv-FC-NC vs. Sham; #$P < 0.05$ Lv-FD vs. MI; $n = 6$ mice for each group. (Mean ± SEM; ANOVA followed by Dunnett's test for multiple group comparisons, Student $t$ test for two group comparisons)

nylon suture at 2 mm below the border between left atrium and ventricle to create MI. Myocardial ischemia was confirmed by significant elevation of S–T segment in electrocardiograph (ECG). The Sham-operated mice for control underwent the same experimental procedures as the MI group but without ligation of LAD. The mice were sacrificed by cervical dislocation at 12 h after MI.

**Echocardiographic analysis**. The mice were anesthetized by intraperitoneal injection of 2,2,2-tribromoethanol (200 mg kg$^{-1}$; Sigma, St. Louis, MO, USA). They were then placed in the supine or lateral position on a warming pad. LV function was assessed by two-dimensional guided M-mode and transthoracic echocardiography with the echocardiographic system equipped with a 10.0-MHz phase-array transducer (Vevo2100, Visualsonics, Canada). Two-dimensional end-diastolic and end-systolic long-axis views of LV were standardized as follows: inclusion of the apex, the posterior papillary muscle, the mitral valve, and the aortic root. LV long-axis and short-axis papillary muscle level M-mode curves were recorded for measuring diastolic LV anterior wall[30], LV internal dimension at systole, thickness of diastolic LV posterior wall, thickness of systolic LV posterior wall, EF, and FS.

**Determination of cardiac CV by optical mapping**. Mice were heparinized and euthanized by 2,2,2-tribromoethanol (200 mg kg$^{-1}$, intraperitoneal injection; Sigma, St Louis, MO, USA). The heart was isolated and Langendorff perfused with Tyrode's solution (NaCl 128.2 mM, CaCl$_2$•2H$_2$O 1.3 mM, KCl 4.7 mM, MgCl$_2$•6H$_2$O 1.85 mM, NaH$_2$PO$_4$•2H$_2$O 1.19 mM, Na$_2$CO$_3$ 20 mM, and glucose 11.1 mM; pH 7.35) at 37 °C. After 10 min of stabilization, the heartbeat was ceased by perfusion with the electric-mechanical uncoupler Blebbistatin (50 nM ml$^{-1}$, Selleckchem, Houston, TX, USA) and paced at a frequency of 10 Hz at apex. The heart was then loaded with 30 μl voltage-sensitive dye (RH 237 [N-(4-Sulfobutyl)-4-(6-(4-(dibutylamino)phenyl)hexatrienyl)pyridinium, inner salt]; AAT Bioquest Inc., USA). Five min later, the dye was excited at 710 nm using monochromatic light-emitting device[31,32]. Images were acquired with a MiCAM05 CMOS camera (SciMedia, USA) at 2000 frames s$^{-1}$ with a filter setting of 1 kHz. Cardiac CV was calculated from the gradient of the scalar field of 12-ms isochronal activation maps along the septal apex–base axis. The time for a cardiac excitation to travel a certain distance was measured, and CV was calculated by the equation: CV = distance/conduction time.

**Determination of cardiac CV by surface ECG analysis**. The apex to base conduction was measured as the longitudinal conduction of the heart, which is presumably mediated by gap junction channels. Mice after varying treatments were anesthetized 2,2,2-tribromoethanol (200 mg kg$^{-1}$, i.p.). Hearts were then quickly excised and placed in ice-cold bicarbonate-buffered Krebs–Henseleit solution (mM:

NaCl 119, NaHCO$_3$ 25, KCl 4, KH$_2$PO$_4$ 1.2, MgCl$_2$ 1, CaCl$_2$ 1.8, glucose 10, and Na-pyruvate 2; pH adjusted to 7.4) bubbled with 95% O$_2$/5% CO$_2$ (Liming Oxygen Company, Harbin, China)[33]. The heart was washed to clean up the blood and quickly mounted to the Langendorff perfusion apparatus. Two pairs of recording electrodes were fixed at the apex and base of the heart, respectively, and connected to the BL-420E$^+$ integrated signal acquisition and processing system (Techman Software Company, Chengdu, China). Activations of cardiac muscles at the sites of apex and base were constantly recorded. The distance ($D$) between the two recording electrodes were measured and taken as the length of travel of excitation waves from apex to base. The appearance of the peaks of the second deflections within the activation complex on the surface electrocardiograms was set as the onset of cardiac activation at the recording site. The lag time ($\Delta T$) between cardiac activations at apex and base represents the time required for an excitation wave to travel the distance ($D$) between the two sites. The cardiac CV was then calculated according to the equation: CV = $D\Delta T^{-1}$. For each data point of CV, 20 consecutive cardiac activations were analyzed, and the final values were averaged from the 20 measurements. The CV measured in such a way represents the longitudinal conduction of the heart, which is presumably mediated by gap junction channels.

**Assessment of propensity of ventricular arrhythmias**. C57BL/6 mice were anesthetized with 2,2,2-tribromoethanol (200 mg kg$^{-1}$, i.p.). An eight-electrode catheter (1.1F, Octapolar EP catheter; SciSense Inc., London, UK) was inserted through the jugular vein into the right ventricle of mice after anesthesia with the procedures essentially the same as detailed in our previous study[34]. Intracardiac pacing was performed through the catheter electrodes using an automated stimulator interfaced with the data acquisition system (GY6000; HeNan HuaNan Medical Science & Technology Ltd., Zhengzhou, China), and ventricular activation was recorded directly from the surface recording electrode fixed on LV epicardium. Inducibility of VT was determined by applying a train of ten consecutive electrical pulses with a coupling interval of 80 ms (S1), followed by two extra stimuli (S2 and S3) at coupling intervals of 2 ms, respectively. Successful induction of VT was defined as the appearance of rapid nonsinus rhythm ventricular activations lasting for three beats or more.

**RNA extraction**. Total RNA was extracted from LV tissues from mice or human subjects using Trizol reagent according to the manufacturer's instructions (Invitrogen, Camarillo, CA, USA) followed by phenol/chloroform extraction. The quality of the extracted RNA samples was confirmed by denaturing gel showing discrete 28 s and 5 s bands without smear.

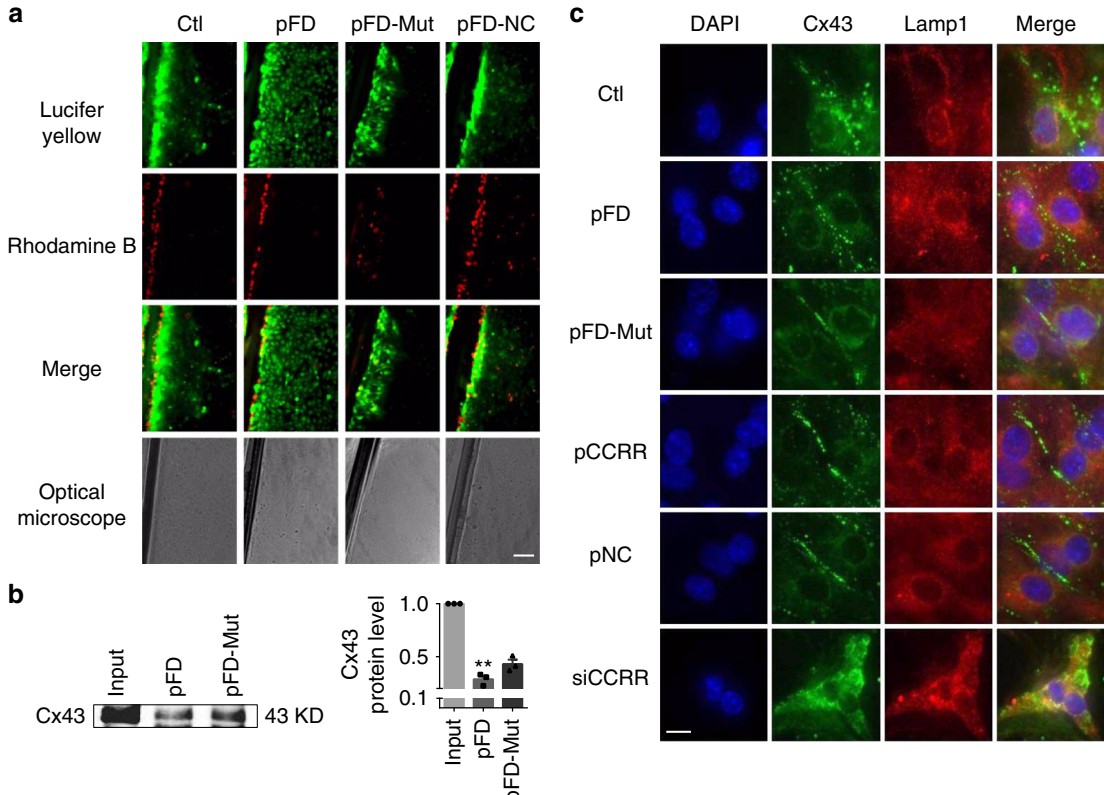

**Fig. 9** Functional domain of CCRR (FD) improves gap junction communication. **a** Effects of conserved FD on gap junction communication assessed by dye transfer techniques in AC16 human adult ventricular cells (200× magnification). AC16 cells were loaded with a diffusible fluorescent dye (Lucifer yellow), and the diffusion of the dye was monitored under a laser confocal microscope. Note that transfection of pFD for FD overexpression significantly promoted, whereas pFD-Mut did not alter, the diffusion of Lucifer yellow among AC16 cells (green). The high molecular weight marker dye conjugate Rhodamine B was used as a negative control (red). Similar data were acquired from another three experiments. **b** Effect of FD on the interaction between Cx43 and CIP85, as reported by co-immunoprecipitation in cultured NMVMs transfected with the plasmid carrying the FD (pFD) for overexpression. Note that the protein level of Cx43 co-immunoprecipitated by CIP85 antibody was significantly reduced. pFD-Mut: the mutant construct of pFD with nucleotide substitution to the putative binding site of CCRR or FD to CIP85. **$P < 0.01$ pFD vs. pFD-Mut; $n = 3$ batches of cells for each group. (Mean ± SEM; ANOVA followed by Dunnett's test for multiple group comparisons, and Student $t$ test for two group comparisons). **c** Effect of CCRR or FD on the recruitment of Cx43 to lysosome and the presence of Cx43 in the plasma membrane determined by co-immunostaining of Cx43 and lysosome marker Lamp1 in cultured NMVMs (600× magnification). Note that either pCCRR or pFD weakened the overlapping staining of Cx43 and Lamp1, and enhanced the Cx43 staining to the cytoplasmic membrane. $n = 3$ batches of cells for each group

**Quantitative real-time RT-PCR (qPCR).** Complementary DNA synthesis was performed with random primer according to the manufacturer's instructions (Reverse Transcription System, Promega; Cat#1202162). The SYBR Green PCR Master Mix Kit (Applied Biosystems; Cat#4309155) was used for real-time PCR to quantify the target genes on the ABI 7500 fast Real-Time PCR system (Applied Biosystems, USA). The specificity of the amplified product was monitored by its melting curve. PCR primers were designed using the Primer Premier 3.0 program (PREMIER Biosoft International, USA). Relative expressions of lncRNAs were calculated using the comparative cycle threshold (Ct) method ($2^{-\Delta\Delta Ct}$). Each data point was then normalized to GAPDH as an internal control in each sample. The final results are expressed as fold changes by normalizing the data to the values from control subjects.

**Isolation of neonatal mouse ventricular myocytes (NMVMs).** Cardiomyocytes were isolated from 1-day-old neonatal mice[19], neonatal mouse ventricles were cut into 1–2 mm³ after the hearts had been rapidly removed, and cells dissociated in 0.25% trypsin at 37 °C for 10 min. Heart tissues were trypsinized until the tissues disappeared and cell suspensions were collected by centrifugation at 1500 g for 5 min. The collected cells were then resuspended in DMEM (Biological Industries, Kibbutz Beit Haemek, Israel) supplemented with 10% fetal bovine serum (Biological Industries, Kibbutz Beit Haemek, Israel) and penicillin (100 U ml⁻¹)/streptomycin (100 U ml⁻¹; Beyotime, Shanghai, China), and cultured at 37 °C in 5% $CO_2$ and 95% air in a humidified incubator. After 90 min for fibroblast adherence, the cell suspension was plated into 6-well plate at $3 \times 10^5$ cells per well with DMEM. 5-bromo-2-deoxyuridine (10 nM) was added into the medium to remove fibroblasts.

**AC16 cell culture.** AC16 human adult ventricular cardiomyocyte cell line was a gift from Dr. Dongmei Zhang from Dalian Medical University. The cells were maintained in DMEM F-12 supplemented with 10% fetal bovine serum.

**Assessment of gap junction communication by the dye transfer method.** After varying treatments as specified in the main text, confluent AC16 human cardiomyocytes seeded on the coverslips were washed three times with 37 °C PBS. They were then plated into 35-mm dishes containing DMEM F-12 cell culture medium (Invitrogen Gibco, Carlsbad, CA, USA). Subsequently, 2 ml PBS containing the low molecular weight gap junction-permeable fluorescent dye Lucifer yellow (1%; Sigma, St. Louis, MA, USA) and the high molecular weight marker dye conjugate Rhodamine B dextran (1%; Sigma, St. Louis, MA, USA) was added to the center region of the coverslip. A 27-gauge needle was used to create longitudinal scratch through the cell monolayer. The cultures were rinsed with PBS 1 min following dye loading. The diffusion of the fluorescent dye was monitored under a laser confocal microscope and the distance of dye diffusion from the scratch margin was measured.

**Western blot analysis.** The total protein, surface membrane protein, and cytoplasmic protein samples were extracted from cardiac tissues of C57BL/6 mice or primary cultured cardiomyocytes for immunoblotting analysis. For the total protein analysis, frozen tissue was homogenized with 1000 μl solution contained 40% sodium dodecylsulfate (SDS), 60% RIPA and 1% protease inhibitor (Cat#539131; Millipore, MA, USA) in each 200 mg cardiac tissue. The homogenate was then centrifuged at 12,000 g for 30 min and the supernatants (containing cytosolic and membrane fractions) were collected. Extraction of surface and cytoplasmic proteins

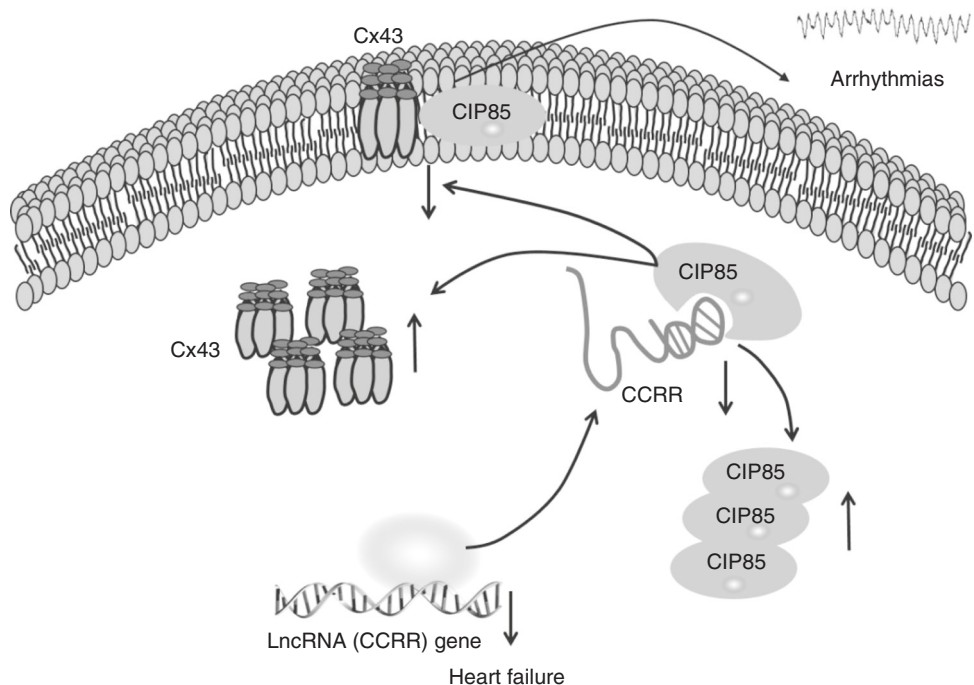

**Fig. 10** Schematic illustration of the proposed signaling pathway for CCRR actions. CCRR binds to CIP85 to prevent the binding of CIP85 to Cx43 to reduce the internalization of Cx43 and block CIP85-induced endocytic trafficking of Cx43 thereby rescuing the functional presence of Cx43 in the cytoplasmic membrane

was conducted using the Surface and Cytoplasmic Protein Reagent Kit (Cat#P0033; Beyotime, Shanghai, China) according to the manufacturer's instructions. Protein concentrations were measured spectrophotometrically using the BCA kit (Universal Microplate Spectrophotometer; Bio-Tek Instruments, Winooski, VT, USA). Protein samples were fractionated by SDS–polyacrylamide gel electrophoresis (10% polyacrylamide gels for Cx43) then transferred to PVDF membrane. The primary anti-Cx43 (Cat#ab11370, 1:5000 dilution; Abcam, Cambridge, UK) and anti-CIP85 antibodies (Cat#sc-86867, 1:200; Santa Cruz Biotechnology, CA) were used. GAPDH (Cat#TA-08, 1:1000, Beijing, China) was selected as an internal control for total and cytoplasmic proteins, and anti-N-cadherin antibody (Cat#ab76011, 1:5000; Abcam, Cambridge, UK) was used as an internal control for membrane protein samples. Western blot bands were captured on the Odyssey Infrared Imaging System (LI-COR Biosciences, Lincoln, NE, USA) and quantified with Odyssey v1.2 software by measuring the band intensity (area × OD) in each group and normalizing to the internal control. All antibodies were diluted in PBS buffer. The uncropped scans of the most important blots are provided in Supplementary Figure 13.

**Northern blot analysis**. Northern blot was performed using the siphon method. The DIG Northern Starter Kit (Roche, Indianapolis, USA) was used. According to the manufacture's instruction, 15 μg RNA was used and the electrophoresis was employed by 1% formaldehyde denaturing gel, and then transferred to the membrane by siphoning. Digoxygen-labeled probes were used to incubate cation-exchanged nylon membranes, ECL was developed for 5 min, and scanning imaging was performed. The sequence of probe: Forward: TAATACGACTCACTA-TAGGAACCATTGCAGGACTGAACCG, Reverse: ATAGGCAGAGCGGACTG TGA.

**Immunohistochemistry**. For mouse tissue, animals were anesthetized with 2,2,2-tribromoethanol (200 mg kg$^{-1}$, i.p.; Sigma, St. Louis, MO, USA), and the hearts were quickly removed and perfused transcardially with 4% buffered PFA (pH 7.4). For human cardiac tissues, the preparations were frozen and stored in liquid nitrogen before use. The cardiac tissues were removed, dehydrated and frozen in OCT (opti-mum cutting temperature compound), and 6-μm sections were mounted on glass slides. After blocking, sections were incubated with the primary anti-α-actinin antibody (Cat#A7811, 1:200; Sigma, Saint Louis, USA), anti-Cx43 (Cat#ab11370,1:2000; Abcam, Cambridge, UK), anti-Cx43 (Cat#ab11369, 1:1000; Abcam, Cambridge, UK), or anti-CIP85 antibodies (Cat#sc-86867, 1:100; Santa Cruz Biotechnology, CA) at 4 °C overnight. The cardiac slices were then washed and incubated with the secondary antibodies conjugated to Alexa Fluor 488 (1:200) and Alexa Fluor 594 (1:200) (Invitrogen, Camarillo, CA, USA) at room temperature for 1 h. All antibodies were diluted in PBS buffer. The slices were examined under a laser confocal microscope.

**Immunocytochemistry**. This experiment was conducted to visualize changes of expression and subcellular distribution of Cx43 and CIP85 proteins and Lamp1 in cultured NMVMs. Cultured NMVCs were incubated with the anti-connexin-43 antibody (Cat# ab11369, 1:1000; Abcam, Cambridge, UK), or anti-CIP85 antibody (Cat# sc-86867, 1:100; Santa Cruz Biotechnology, CA, USA) or anti-Lamp1 antibody (GT25212, 1:500; GeneTex, USA) at 4 °C overnight. The cells were washed and incubated with the secondary antibodies conjugated to Alexa Fluor 488 (1:200) and Alexa Fluor 594 (1:200) (Molecular Probes, Eugene, OR, USA) at room temperature for 1 h. The preparations were then examined under a laser confocal microscope.

**Fluorescence in situ hybridization (FISH)**. To visualize the cellular distribution and expression alteration of CCRR, FISH was performed on the frozen sections of LV wall and cultured NMVCs with a CY3 specific probe for CCRR (5′-cgctttccctggaaaagctg-CY3) or a negative control probe without CY3. The frozen section was fixed with 4% paraformaldehyde for 20 min, washed three times with PBS every 5 min, treated with 0.2 N HCl at room temperature for 5 min, and then incubated with acetic anhydride and triethanolamine solution at room temperature for 10 min. The samples were prehybridized at 37 °C for 2 h, crosslinked at 37 °C overnight, washed twice with 5× saline sodium citrate (SSC), and washed three times with 50% diamide/2× SSC at 50 °C. DAPI (Roche, Indianapolis, USA) was used to stain nuclei, and fluorescence resistance agent was used to avoid quenching.

**Construction of lentivirus vectors**. A lentivirus vector carrying the interference fragment to CCRR/AK045950 (Lv-siCCRR) for loss-of-function and the CCRR overexpression vector-GV367 containing the full-length CCRR/AK045950 sequence (Lv-CCRR) for gain-of-function and conserved FD of CCRR (Lv-FD) were constructed by Shanghai GeneChem Co., Ltd. (Shanghai, China). The PDONR221 carrier vector was used to complete BP recombination reaction with 14MR0004-03 in order to obtain an entry vector containing the interference sequence. The interference sequences were as following: Forward: TGCTGTTT AGAAGGTGACTCCA GGAAGTTTTGGGCCACTGACTGACTTCCTGGACACC TTCTAAA; Reverse: CCTGT TTAGAAGGTGTCCAGGAAGTCAGTCAGTGG CCAAAACTTCCTGGAGTCACCTTCTAAAC. The LR recombination reaction was carried out on the entry vector and the slow virus target vector pLenti6.3/V5-DEST in order to obtain the slow virus expression vector containing the interfering sequence. The vector was identified after analyzing the plasmid sequence (Invitrogen, Camarillo, CA, USA). The successful construction of the vector was verified by restriction enzyme digestion, PCR identification and sequencing. The constructed plasmid and lentivirus packaging plasmids phelper 1.0 and phelper 2.0 were co-transfected into the cultured HEK293T cells. Lentiviral particles were obtained by collecting supernatant using the kit for ultracentrifugation concentration and purification of lentiviral particles and combined with fluorescent titer assay and ELISA. Virus titer was determined as $1 \times 10^8$ transducing U ml$^{-1}$.

The viral constructs were administered into mouse heart by intracavity injection (injected into left ventricular chamber) after open heart surgery 1 week before TAC procedures.

**Transfection of lncRNA siRNA and plasmids**. LncRNA siRNA for CCRR (siCCRR) and a scrambled negative control RNA (siNC) were synthesized by Invitrogen (Carlsbad, CA, USA), and siRNA for CIP85 (siCIP85) by Shanghai Gene Pharma Co., Ltd. (Shanghai, China). These siRNAs were transfected at a final concentration of 100 nM or 2 μg plasmids (pCCRR, pFD, pFD-Mut) into NMVMs using the X-treme GENE Transfection Reagent (Roche, Indianapolis, USA) according to the manufacturer's protocols.

**CIP85–Cx43 co-immunoprecipitation**. We followed essentially the procedures established by Cochrane et al.[22] with minor modifications. Briefly, NMVMs were lysed in RIPA lysis buffer (50 mM Tris/pH 7.4, 150 mM NaCl, 1% Triton X-100, 1% sodium deoxycholate, 0.1% SDS, sodium orthovanadate, sodium fluoride, EDTA, leupeptin, and protease inhibitors) supplemented with protease inhibitors and the lysate was immunoprecipitated with a monoclonal Flag antibody (Sigma, USA) at 4 °C. As a negative control, an equal amount of the lysate was immunoprecipitated with a monoclonal GST antibody (Santa Cruz Biotechnology, CA, USA). The immune complexes were collected with protein A/G plus-agarose beads (Santa Cruz Biotechnology, CA) and washed ten times with PBS. Flag-CIP85 and Cx43 were detected by immunoblotting. The physical interaction between endogenous CIP85 and Cx43 was explored by precipitating with the CIP85 antibody or a polyclonal Flag antibody (Santa Cruz Biotechnology, CA, USA) as a negative control.

**RIP immunoprecipitation**. RNA immunoprecipitation was performed using the Magna RIP™ RNA-Binding Protein Immunoprecipitation Kit (Millipore, Darmstadt, Germany) according to manufacturer's instructions, followed by real-time RT-PCR using the SYBR Green PCR Master Mix Kit (Applied Biosystems, Carlsbad, CA). RNAs were in vitro transcribed using the T7 RNA Polymerase Kit (Beyotime, Shanghai, China), The gene-specific primers sequences were as following: CCRR (mouse), Forward: GACTGAGCTTTGAAAATATG; Reverse: GTCCCATCCCCAAGCTGCTTGATC. FD (mouse), Forward: GGGAGATAAT CACGTTCTGTT; Reverse: TCTGTCGGGACACTCATCT. The Pierce™ Magnetic RNA–Protein Pull-Down Kit (Thermo, Rockford, USA) was used in RNA–Protein pulldown experiments according to manufacturer's instructions. The proteins pulled down by CCRR, FD of CCRR, zinc finger antisense 1 (*ZFAS1*, an lncRNA as another negative control), antisense of CCRR (AS, as a negative control), or positive RNA control androgen receptor (AR) were detected by Western blotting with the anti-CIP85 antibody (NBP2-60726, Novusbio, CA, USA). AR RNA, the proximal 3′untranslated-region of AR (5′-CUGGGCUUUUUUUUUCU CUUUC UCU CCUUUCUUU UUCUUCUUCCCUCCCUA-3′) was used as a positive control for the pulldown assay, and it contains UC-rich regions for HuR (RNA-binding protein; Cat #12582S, Cell Signaling Technology, USA).

**Data analysis**. Experimental data are expressed as mean ± SEM and they were analyzed with SPSS 24.0 software. Statistical comparisons among multiple groups were performed using analysis of variance (ANOVA) followed by Dunnett's test. Student $t$ test was carried out for comparisons between two groups. $\chi^2$-test was used for nonparametric data set comparisons. A two-tailed $P < 0.05$ was taken to indicate a statistically significant difference. The sample size for whole animal experiments was set to be >5 mice for each group, and for molecular biology experiments, the sample size for set to 3–6 for each group and each data point was repeated in triplicate. The data on the animals died before the completion of the whole experimental procedures were excluded from our data analysis.

**Code availability**. No computer code was used to generate results for the current study.

## Data availability

CCRR sequence data that support the findings of this study have been deposited in GenBank with the primary accession codes "AK045950.1" (https://www.ncbi.nlm.nih.gov/nuccore/ak045950).

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

## Acknowledgments

This work was supported in part by the grants from the Funds for Creative Research Groups of the National Natural Science Foundation of China (81421063), National Key Research and Development Program of China (2017YFC1307403), the Key Project of National Science Foundation of China (81530010), and the Natural Science Foundation of China (81570357, 81371211, 81570301, 81770284, 81773735, and 81673424).

## Author contributions

L.H.S., Y.Z., L.N.X., H.L.S. and B.F.Y. designed, performed experiments, and supervised all aspects of the research and analyses. L.H.S., H.L.S. and B.F.Y. wrote and finalized the manuscript. L.H.S., Y.Z., L.N.X., Z.W.P., X.L.H., H.Q.R., J.Y.Z., L.J., Z.G.L., L.N.C., X.X. W., T.T.Y., S.Q.W., B.B.F., Y.G. and C.Y. participated in research, data analysis and interpretation. H.Y.L., Z.H.L. and W.X.M. were responsible for collecting heart tissue samples. Y.L.B., Y.Z., C.Q.X., Z.G.W., X.Y.Z. and Y.J.L. contributed to discussing, correcting, and finalizing the manuscript.

## Additional information

**Competing interests:** The authors declare no competing interests.

