## [Peer Review File · Nature Communications]

Reviewers' comments:

Reviewer #1 (expert in cardiac physiology and connexin-43)

Remarks to the Author:

This manuscript describes studies to identify and characterize a long non-coding RNA-dependent mechanism implicated in conduction slowing and contractile abnormalities in a model of pressure-overload heart failure. The authors first identified a lnc RNA that was downregulated in heart failure they renamed CCRR, and then, through a series of cellular, murine in vivo and human tissue sample studies, the authors propose the existence of a pathway through which heart failure (HF) leads to the activation of NFATc3, which represses CCRR expression, leading to increased accumulation of CIP85, increased endocytic internalization of Cx43, decreased gap junction function and conduction slowing, arrhythmogenesis and contractile deficits.

If shown to be true, this would represent an interesting, novel and potentially "targetable" pathway to lessen the disease manifestations associated with heart failure. That being said, there are a number of questions associated with experimental methodology and interpretation that require some clarification.

There are some issues with quantification of results. In many figures, using a range of techniques (FISH, Western blotting, immunostaining, etc.), the representative example shown seems significantly less impressive than the summary bar-graph data, or in some cases, the images are just not of sufficient resolution to be sure either way. By way of example, the degree of downregulation of CCRR in the HF tissue shown in Figure 1C is not particularly impressive, especially if downregulation of CCRR is critical for the validity of the entire pathway being proposed. Also, why do the authors believe CCRR should be organized into plaques?

Another example is the Western blot in Figure 3C. The increase of Cx43 in the membrane shown after transduction with Lv-CCRR seems no greater than that seen with the Lv-NC negative control. The companion immunofluorescence in Figure 3E has the red channel so bright that it is very difficult to evaluate the Merged images showing Cx43 expression and its localization and abundance.

Other issues:

Why was intra-cavity infection used in some experiments and intra-coronary infection used in others? There is no explanation offered.

For some of the Western blots, the number of samples for membrane fraction is 3 but for cytoplasmic it is 5. Why?

The epicardial recordings to measure conduction velocity involve single electrograms taken ~ 1cm apart. This is not very accurate, especially in the case of control hearts versus failing (and presumably enlarged hearts). Multi-electrode recordings or better yet, optical mapping with voltage sensitive dyes would be much more accurate. Related to this, how are you determining the moment of activation? In other words, how did you determine where to place the blue vertical line for each tracing? It does not seem consistent. Are these measurements done in sinus rhythm? It appears to be the case but please confirm.

The Lv-NC seems to have shortened the VT Duration parameter almost as much as Lv-CCRR? Please comment.

The contractile deficit observed in this model is not seen with Cx43 KO, suggesting other factors

besides Cx43 downregulation contribute to this aspect of the phenotype. Please comment. Its also unclear how restoration of Cx43 expression by transduction with CCRR so potently improves contractile function.

The authors measure Nav1.5 levels and find them to be increased in the HF model, which is perhaps a bit surprising. Moreover, knocking down CCRR with Lv-siCCRR increases Nav1.5 levels. This contrasts with Cx43, in which lower levels of CCRR are associated with decreased Cx43. How do the authors reconcile this major difference?

Nav1.5 levels may be increased, but to equate the level of protein expression with a functional parameter, i.e. "intracellular conduction" – is not justified. There are no data indicating functionality of the sodium channels.

Non-quantitative terms like "an impressive quantity..." probably are not the best to use.

The manuscript will require careful editing to improve grammar.

Reviewer #2 (expert in cardiac physiology and connexin-43)

Remarks to the Author:

This manuscript presents the intriguing possibility that lncRNA AK045950 ("CCRR*") regulates Cx43 trafficking and is reduced in heart failure, decreasing Cx43 based cell-cell coupling and increasing arrhythmogenesis. The evidence of CCRR reduction in whole hearts and of intercalated disc structural changes, reduced EF, as well as the evidence of arrhythmia induction with reduced CCRR are convincing. However the models used, mechanism of arrhythmogenesis related to Cx43 trafficking, and explanation why CCRR reduces ejection fraction as well as increasing arrhythmia, all require significantly more clarification. The data suggest that CCRR relates to structural proteins of cardiomyocytes or other cells that influence cardiomyocytes, important for heart function as well as intercalated disc maintenance, rather than primarily targeting gap junctions as the name CCRR suggests.

1. FISH data in mouse cardiac muscle (Figure 1b) are statistically significant but quantitatively small and, based on the representative images, can be confounded by a high background signal. The modest reductions in myocytes in the setting of large reductions in mouse and human hearts suggest a non-cardiomyocyte contribution.
2. Reduced EF and disruption of the intercalated disc with rescue by CCRR are evidence of a structural consequence of CCRR rather than electrical. Cx43 gap junctions are likely the final read-out of an intact disc and cytoskeleton delivery apparatus. It is quite likely that CCRR does not directly affect Cx43 protein but rather disc structure which then can affect both Cx43 delivery and internalization.
3. CIP85 changes are modest at best and appear too low to explain significant changes in Cx43 distribution. Other key factors have been identified with dramatic effects on Cx43 internalization such as phosphorylation status of casein kinase (CK) serines as well as serine 373 and 368. It is not clear why the investigation was restricted to CIP85. It is actually not clear why, assuming a primary defect in Cx43 trafficking, the investigation assumed the trafficking affect was internalization and not forward delivery of Cx43.
4. The definition of HF, reduced EF, used for the TAC mice is invalid. HF is a clinical syndrome of cardiac origin volume overload. Reduced EF (< 50%) suggests the onset of dilated cardiomyopathy but does not provide evidence that the mice have HF. Body weight, lung weight, and physical activity measures are typically used to establish HF.

5. The human samples of HF have considerable variability. Several of the samples of preserved ejection fraction while others have reduced ejection fraction. Two of the patients had mitral stenosis which could result in heart failure yet with an unaffected ventricle. A uniform reduced of CCRR in these samples argues against HF induced reductions and may be due either to other forms of cardiac disease or, quite possibly, an artifact of obtained samples from deceased individuals.

Reviewer #3 (expert in cardiac physiology and lncRNA)

Remarks to the Author:

In this report, Sun et al, reported the functional characterization of an lncRNA, termed CCRR in cardiac conduction regulation. The expression of CCRR was measured in heart failure tissue. Loss- and gain-of function studies in mouse hearts and cultured myocytes were conducted to evaluate the impact of CCRR. In addition to slowing electric conduction, arrhythmogenesis and ultrastructure, changes in cardiac function and gene expression were also observed. Furthermore, CCRR impact on Cx43/CIP85 expression and distribution, as well as its interaction with Cx43/CIP85 complex was characterized. Finally, NFAT3 mediated regulation of CCRR down-regulation was demonstrated.

The concept of an lncRNA mediated modulation of cardiac conduction via direct interaction with Cx43/CIP85 complex is novel and potentially important based on in vivo gain/loss-of-function analysis. However, the current report appears to be quite preliminary in developing this hypothesis and in revealing the underlying mechanisms. The validity of the central conclusion should also be better demonstrated.

1). CCRR connection with abnormal EP in heart failure. Authors indicated that informatic analysis based on co-expression network of lncRNAs and coding RNAs in failing heart led to the initial hypothesis that CCRR was functionally linked to cardiac conduction. However, no specific details were provided to demonstrate how such connection was established and how significant of this connection from informatics analysis. In fact, both data from gain and loss of function studies appeared to indicate CCRR had a broad impact on cardiac pathology in addition to conduction defects, such as contractile dysfunction.

2). CCRR expression: Authors used both RT-PCR and FISH to detect CCRR expression. For lncRNA, it is important to validate the expression of full length transcript or other potential isoforms by northern blots. In addition, the FISH analysis in Figure 1 was not adequate in quality and amplifications, and lacked positive and negative controls. It is not clear the cellular structure associated with the enriched staining. But the pattern does not fit cell-cell junction or correlated with localization of Cx43/CIP85. This is a major concern.

3). Function of CCRR:

A. authors presented evidence to support the effect of over-expression or siRNA KD on mouse heart function and cardiac conduction. As shown in Supplemental Figure 2, the KD effect was very modest comparing to the observed reduction in HF samples. It is also not clear about the duration of CCRR over-expression throughout the development of HF (it is not clear at what time points during the experiment the data in Supplemental Figure 2 was obtained). In addition, the KO or the overexpression by lentiviral vectors were performed without targeted cell-type specificity.

Therefore, it is not clear in these experiments, the observed changes in cardiac phenotype were entirely due to their effect in cardiomyocytes. In addition to inter-cellular coupling, interstitial fibrosis can be a major contributor to conduction defects and arrhythmia. Authors observed significant changes in cardiac function in addition to cardiac conduction or rhythm. These changes may be related to each other. However, without more detailed temporal analysis and targeted intervention, it is not clear if cardiac conduction defect is a direct and specific outcome of CCRR expression or indirectly through contractile dysfunction.

B. The specific impact of CCRR expression on cardiomyocyte inter-cellular conduction should be demonstrated directly in cultured myocytes based on the effect of its gain/loss of CCRR expression

on Cx43/CIP85 expression or distribution and cell-cell junction (by dye transfer). This would be also a suitable system to interrogate the mechanisms.

4). Mechanism of CCRR function: Authors provided intriguing evidence that CCRR interacted with CIP85 directly and regulated Cx43 distribution at gap junction via CIP85/Cx43 complex. However, there are several significant issues with the supporting data.

A. The data shown in Figure 3C/D is puzzling regarding the intracellular distribution. Cx43 is synthesized and inserted into the ER membrane, and then transported to cytoplasmic membrane followed by recycling back into lysosome for degradation or reinsertion. In heart, majority of Cx43 protein signaling should be detected on membrane. Authors should provide more details about the conditions used for fractionation (detergent content) and additional data to support the quality of the fractionation method. The same concern applies to CIP85, except in this case, the immunostaining shown in Figure 3e did not support discrete pattern of CIP85 distribution. The colocalization with Cx43 was clearly over-interpreted based on the overly saturated CIP85 signal covering the entire myocyte.

B. Authors stated that CCRR regulates Cx43 via direct interaction with CIP85. However, the pull-down recovery with CIP85 for CCRR was very low (comparing with the total input), and there is no good control to support the specificity other than IgG. At least, authors should try to use another non-specific lncRNA as a negative control. Previous report from Lau's group indicate that CIP85 expression promotes Cx43 degradation. However, there is no data from current study to support that CIP85 interaction with CCRR had a significant impact on its binding or regulation to Cx43 distribution/cycling. Data shown in Figure 4a and b should be tested under control, HF and with CCRR manipulations. The overall hypothesis that CCRR down-regulation led to enhanced CIP85 interaction with Cx43 and its internalization and degradation lacks clear evidence and remains speculative at this time.

5. NFAT mediated regulation of CCRR down-regulation. This is an interesting observation as NFAT is usually viewed as a transcription activator but the current data only demonstrated NFAT binding to CCRR promoter regions and its expression is correlated with CCRR expression. This is not a strong evidence to support NFAT is an immediate upstream regulator of CCRR expression. Nevertheless, this is not a critical issue for this report and authors should remove it to enhance the focus of this study, i.e. the functional relevance of CCRR in cardiac pathology, particularly, in arrhythmia.

Responses to Reviewers' Comments

Reviewer #1 (expert in cardiac physiology and connexin-43)

Remarks to the Author:

This manuscript describes studies to identify and characterize a long non-coding RNA-dependent mechanism implicated in conduction slowing and contractile abnormalities in a model of pressure-overload heart failure. The authors first identified a lnc RNA that was downregulated in heart failure they renamed CCRR, and then, through a series of cellular, murine in vivo and human tissue sample studies, the authors propose the existence of a pathway through which heart failure (HF) leads to the activation of NFATc3, which represses CCRR expression, leading to increased accumulation of CIP85, increased endocytic internalization of Cx43, decreased gap junction function and conduction slowing, arrhythmogenesis and contractile deficits.

If shown to be true, this would represent an interesting, novel and potentially “targetable” pathway to lessen the disease manifestations associated with heart failure. That being said, there are a number of questions associated with experimental methodology and interpretation that require some clarification.

Answer: We thank you for your very positive comments on our work. Our point-by-point explanations on the issues you raised are listed below.

There are some issues with quantification of results. In many figures, using a range of techniques (FISH, Western blotting, immunostaining, etc.), the representative example shown seems significantly less impressive than the summary bar-graph data, or in some cases, the images are just not of sufficient resolution to be sure either way. By way of example, the degree of downregulation of CCRR in the HF tissue shown in Figure 1C is not particularly impressive, especially if downregulation of CCRR is critical for the validity of the entire pathway being proposed. Also, why do the authors believe CCRR should be organized into plaques?

Answer: We have double-checked our images and the quantified data based on the images to ensure the consistency between images and corresponding values as well as the accuracy of the data measurements and calculation. We have also invited two non-experimenters to repeat the analyses in a blind manner. The results reassured us that the bar-chart values now truly reflect the images in our study despite of the discrepancies between the visual effect and the values. Nonetheless, in order to address your concerns, we have conducted additional experiments on FISH, Western blotting and immunostaining analyses. The new data are consistent with our original results, and are incorporated into our revised manuscript.

Specifically for the FISH results in Figure 1c, we have to admit that our original analysis was performed under a relatively high brightness or background noise that could likely have significantly contaminated the true signals. We have therefore re-analyzed the images after having readjusted the

brightness to minimize the background. In this way, we are now having a more discrete view of the staining and improved values measured based upon the brightness-adjusted images. As depicted in the new Figure 1c, the CCRR is overall stained evenly throughout the cytoplasm with some rod- or patch-shaped structures that we originally described as “plaques”. We have now removed such a description, despite that with higher magnifications such aggregates could be seen.

Another example is the Western blot in Figure 3C. The increase of Cx43 in the membrane shown after transduction with Lv-CCRR seems no greater than that seen with the Lv-NC negative control. The companion immunofluorescence in Figure 3E has the red channel so bright that its very difficult to evaluate the Merged images showing Cx43 expression and its localization and abundance.

Answer: We thank you for bringing up these issues that we had apparently overlooked. For the Western blot results, we have re-checked our data and found that the main reason for the non-optimized Lv-NC negative control is due to the relatively large variations. To tackle this problem and other related issues as well, we have conducted additional Western blot experiments. With increased the number of experiments, the variations were reduced and the new data have been incorporated into Figure 3c.

For the immunostaining images, we have readjusted the brightness to minimize the background noise so that the discrete localization and abundance of Cx43 could be clearly viewed. The new data are shown in Figure 3e, which reaffirmed our original results and conclusions, and have also improved the quality of our data.

Other issues:

Why was intra-cavity infection used in some experiments and intra-coronary infection used in others? There is no explanation offered.

Answer: We thank you for picking up this issue; our original description is inaccurate. We have used only intra-cavity injection (injected into left ventricular chamber) throughout the experiments. Such a method has been well believed to yield circulation of viral vectors through coronary artery to enhance the delivery of the constructs. We have now corrected the description and unambiguously stated “intra-cavity injection” (page 6, line 10 and page 16, lines 9-10).

For some of the Western blots, the number of samples for membrane fraction is 3 but for cytoplasmic it is 5. Why?

Answer: The yield of protein extraction from membrane fraction is much lower than the cytoplasmic fraction, and in some experiments, the concentrations of the membrane protein samples were too low for Western blot

analysis. This is why the number of membrane fraction was smaller than that of cytoplasm. We have now carried out additional Western blot experiments for all relevant data to increase the number of samples and minimize variations.

The epicardial recordings to measure conduction velocity involve single electrograms taken ~ 1cm apart. This is not very accurate, especially in the case of control hearts versus failing (and presumably enlarged hearts). Multi-electrode recordings or better yet, optical mapping with voltage sensitive dyes would be much more accurate. Related to this, how are you determining the moment of activation? In other words, how did you determine where to place the blue vertical line for each tracing? It does not seem consistent. Are these measurements done in sinus rhythm? It appears to be the case but please confirm.

Answer: We are lucky enough to have received our new instrument (MiCAM05 CMOS camera (SciMedia,USA)) for optical mapping of cardiac activation at the whole-heart level right during the course of our manuscript revision. This instrument allows for more straightforward and accurate measurements of cardiac activation and conduction. We have therefore repeated the measurements of cardiac conduction velocity using this technique. Our new data are in perfect agreement with our original conclusion that CCRR plays a key role in maintaining cardiac conduction and knockdown of CCRR slows cardiac conduction velocity. We therefore have replaced these new results for our data presentation of cardiac conduction in Figure 2, and placed our original data to Supplementary Figure 2 as an additional piece of evidence.

For surface ECG measurements, we used the 2nd deflection of an ECG trace that presumably corresponds to the “R” wave of QRS complex as a time of activation. The time-lag of such activation from different sites was taken as a measurement of conduction. This is a standard method, nothing special.

The Lv-NC seems to have shortened the VT Duration parameter almost as much as Lv-CCRR? Please comment.

Answer: We picked up a wrong trace to show in the original version of our manuscript. We have carefully inspected all traces to ensure the correct grouping, and replaced the wrong trace with the right one in the revised illustration (Figure 2b).

The contractile deficit observed in this model is not seen with Cx43 KO, suggesting other factors besides Cx43 downregulation contribute to this aspect of the phenotype. Please comment. Its also unclear how restoration of Cx43 expression by transduction with CCRR so potently improves contractile function.

Answer: There could be two possible explanations for our observations on the changes of cardiac function. The first explanation is as mentioned by you that CCRR could act on other factors important to cardiac contractile function. However, in the absence of any supporting evidence, this view remains a

speculation. Dedicated studies are required to clarify this issue in the future. We must emphasize that our study does not exclude the possibility that CRR might be able to act on structural proteins of cardiomyocytes or other cells that are important to heart function and intercalated disc maintenance. For this, we have added a brief discussion providing the alternative explanation based on the possible structural alterations for our observations (page 11, lines 5-9).

The second explanation is that the observed changes in cardiac function are due to indirect effects secondary to the actions of CRR on cardiac conduction. That is, targeting gap junctions/Cx43 alone can reasonably justify for the observed damages of intercalated disc and impairment of overall cardiac function. It has long been recognized and accepted that cardiac muscles operate as a single functional syncytium; that is, they contract in unison. Impaired intercellular electromechanical coupling could likely disrupt the functional syncytium resulting in cardiac contractile dysfunction (this point is actually mentioned at the very beginning of our manuscript). Electromechanical coupling between cardiomyocytes is a basic requirement for coordinated mechanical activity in the myocardium. Diminished presence of Cx43 in gap junctions is well known to cause discontinued impulse propagation or electrical uncoupling. This is because that cardiac contractile dysfunction may be accounted for, at least partly, by incoordination in contraction of individual myocytes secondary to fragmentation of the depolarizing wave front [1-3]. In addition, Cx43-associated gap coupling plays a significant role in the regulation of resting Ca^{2+} signaling in normal ventricular myocytes [4, 5], loss of which could hurt contractile function.

While the primary concern about the function of Cx43 has been on cardiac excitation conduction/arrhythmogenesis, a decent number of studies have been publishing demonstrating the role of Cx43 in cardiac contractile function. Indeed, it has been generally believed that downregulation of Cx43 in human failing heart has been implicated not only in arrhythmogenesis but also in contractile dysfunction as well.

For example, early in 1998, Kaprielian *et al* observed that a reduction in gap junction coupling is involved in the pathophysiology of hibernation in patients with ischemic heart disease, possibly by unmasking and exaggerating the existing local inhomogeneities in individual cell excitability, hence disrupting wave-front propagation, slowing conduction, and leading to loss of the local coordination in myocyte contraction [1]. The authors proposed that reduction of Cx43 content in gap junctions may contribute not only to arrhythmogenesis by creating anatomic substrates of abnormal conduction but also to wall motion impairment in hibernating myocardium. In hibernating myocardium, the large Cx43 gap junctions typically found at the periphery of the intercalated disc are smaller in size, and the overall amount of immunodetectable Cx43 per intercalated disc is reduced, compared with normally perfused myocardial regions of the same heart [1]. These findings were the first indication that Cx43 gap junction remodeling contributes to impaired ventricular contraction, in

addition to arrhythmia, in human ischaemic heart disease. Saffitz & Yamada supported the idea and believed that the time has come to broaden our thinking about myocardial gap junction channels beyond their electrical coupling role and to consider their possible roles in coordinating metabolic and contractile functions [2]. Apart from alterations in Cx43 gap junction organization, marked reduction in ventricular Cx43 transcript and protein levels typifies the hearts of transplant patients with end-stage congestive heart failure [3-5]. This Cx43 reduction occurs irrespective of whether heart failure is due to ischemic heart disease, idiopathic dilated cardiomyopathy, or aortic stenosis [6-9]. The reduction in Cx43 is spatially heterogeneous and develops progressively during the course of disease [10]. The average reduction of 50% was observed in the failing human ventricle [8], which is consistent to our finding in the present study.

That heterogeneity in Cx43 expression is critical both to abnormal impulse propagation and contractile dysfunction was elegantly demonstrated experimentally by Gutstein *et al* in chimeric mice formed from Cx43-deficient embryonic stem cells and wild-type recipient blastocysts to model the heterogeneous pattern of gap junction expression typical of diseased myocardium [11, 12]. Such a remodeling of gap junctions has been observed in many forms of heart disease. The consequent loss of synchronous ventricular activation has been hypothesized to result in diminished cardiac performance [11, 12].

Oyamada *et al* found that the expression and localization of Cx43 and gap-junctional intercellular communication are critical for the establishment of a synchronized contraction of cultured neonatal rat cardiac myocytes [13]. The same group further found that gap junctional intercellular communication facilitates synchronized contraction of cardiac myocytes through maintaining synchronous intracellular Ca^{2+} fluctuations [5]. An elegant study documented by Valencik *et al* demonstrated that conditional expression of integrin in mice causes 80% reduction in amplitude of the QRS complex, profound systolic dysfunction, decreased Cx43, loss of gap junctions, and abnormal intercalated discs. Yet, isolated left ventricular myocytes contract normally and exhibit normal Ca^{2+} transients, suggesting that intercell electromechanical coupling has been disrupted [14].

While the above studies demonstrated that loss of Cx43 can render contractile dysfunction, the following experiments show that gain-of-function of Cx43 improves cardiac function. One study found that exogenous Cx43-expressing autologous skeletal myoblasts ameliorate mechanical function and electrical activity of the rabbit heart after experimental infarction [15]. Similar observations were reported by another study: cardiomyocytes can form electromechanical junctions with some skeletal myotubes in co-culture and induce their synchronous contraction via gap junctions [16]. In addition, Lakkisto *et al* observed that the heme oxygenase inducer hemin protects against cardiac dysfunction in ischemic/reperfused rat hearts by a

Cx43-dependent mechanism [17]. Another pharmacological study also claimed the involvement of Cx43 in the protective effects of diltiazem on cardiac function (contraction force) during hypoxic injury [18].

Despite this rational argument, we have added a brief discussion providing the alternative explanation based on the possible structural alterations for our observations (Page 11, lines 5-9).

1. Kaprielian RR, Gunning M, Dupont E, Sheppard MN, Rothery SM, Underwood R, Pennell DJ, Fox K, Pepper J, Poole-Wilson PA, Severs NJ. Downregulation of immunodetectable connexin43 and decreased gap junction size in the pathogenesis of chronic hibernation in the human left ventricle. *Circulation*. 1998;97:651-60.
2. Saffitz JE, Yamada KA. Do alterations in intercellular coupling play a role in cardiac contractile dysfunction? *Circulation*. 1998;97:630-2.
3. Severs NJ, Bruce AF, Dupont E, Rothery S. Remodelling of gap junctions and connexin expression in diseased myocardium. *Cardiovasc Res*. 2008;80:9-19.
4. Li C, Meng Q, Yu X, Jing X, Xu P, Luo D. Regulatory effect of connexin 43 on basal Ca²⁺ signaling in rat ventricular myocytes. *PLoS One*. 2012;7:e36165.
5. Kimura H, Oyamada Y, Ohshika H, Mori M, Oyamada M. Reversible inhibition of gap junctional intercellular communication, synchronous contraction, and synchronism of intracellular Ca²⁺ fluctuation in cultured neonatal rat cardiac myocytes by heptanol. *Exp Cell Res*. 1995;220:348-56.
6. Kostin S, Rieger M, Dammer S, Hein S, Richter M, Klovekorn WP et al. Gap junction remodeling and altered connexin43 expression in the failing human heart. *Mol Cell Biochem* 2003;242:135–144.
7. Kostin S, Dammer S, Hein S, Klovekorn WP, Bauer EP, Schaper J. Connexin 43 expression and distribution in compensated and decompensated cardiac hypertrophy in patients with aortic stenosis. *Cardiovasc Res* 2004;62:426–436.
8. Dupont E, Matsushita T, Kaba R, Vozzi C, Coppen SR, Khan N et al. Altered connexin expression in human congestive heart failure. *J Mol Cell Cardiol* 2001;33:359–371.
9. Kitamura H, Ohnishi Y, Yoshida A, Okajima K, Azumi H, Ishida A et al. Heterogeneous loss of connexin43 protein in nonischemic dilated cardiomyopathy with ventricular tachycardia. *J Cardiovasc Electrophysiol* 2002;13:865–870.
10. Wiegeler RF, van Veen TA, Belterman CN, Schumacher CA, Noorman M, de Bakker JM, Coronel R. Transmural dispersion of refractoriness and conduction velocity is associated with heterogeneously reduced connexin43 in a rabbit model of heart failure. *Heart Rhythm*. 2008;5:1178-85.
11. Gutstein DE, Morley GE, Vaidya D, Liu F, Chen FL, Stuhlmann H, Fishman GI. Heterogeneous expression of gap junction channels in the heart leads to conduction defects and ventricular dysfunction. *Circulation*. 2001;104:1194-9.
12. Gutstein DE, Morley GE, Fishman GI. Conditional gene targeting of connexin43: exploring the consequences of gap junction remodeling in the heart. *Cell Commun Adhes*. 2001;8:345-8.
13. Oyamada M, Kimura H, Oyamada Y, Miyamoto A, Ohshika H, Mori M. The expression, phosphorylation, and localization of connexin 43 and gap-junctional

- intercellular communication during the establishment of a synchronized contraction of cultured neonatal rat cardiac myocytes. *Exp Cell Res.* 1994;212:351-8.
14. Valencik ML, Zhang D, Punske B, Hu P, McDonald JA, Litwin SE. Integrin activation in the heart: a link between electrical and contractile dysfunction? *Circ Res.* 2006;99:1403-10.
 15. Antanavičiūtė I, Ereminienė E, Vysockas V, Račkauskas M, Skipskis V, Rysevaitė K, Treinys R, Benetis R, Jurevičius J, Skeberdis VA. Exogenous connexin43-expressing autologous skeletal myoblasts ameliorate mechanical function and electrical activity of the rabbit heart after experimental infarction. *Int J Exp Pathol.* 2015;96:42-53.
 16. Reinecke H, MacDonald GH, Hauschka SD, Murry CE. Electromechanical coupling between skeletal and cardiac muscle. Implications for infarct repair. *J Cell Biol.* 2000;149:731-40.
 17. Lakkisto P, Csonka C, Fodor G, Bencsik P, Voipio-Pulkki LM, Ferdinandy P, Pulkki K. The heme oxygenase inducer hemin protects against cardiac dysfunction and ventricular fibrillation in ischaemic/reperfused rat hearts: role of connexin 43. *Scand J Clin Lab Invest.* 2009;69:209-18.
 18. Matsushita S, Kurihara H, Watanabe M, Okada T, Sakai T, Amano A. Inhibition of connexin43 dephosphorylation is involved in protective effects of diltiazem on cardiac function during hypoxic injury. *Histol Histopathol.* 2011;26:315-22.

The authors measure Nav1.5 levels and find them to be increased in the HF model, which is perhaps a bit surprising. Moreover, knocking down CCRR with Lv-siCCRR increases Nav1.5 levels. This contrasts with Cx43, in which lower levels of CCRR are associated with decreased Cx43. How do the authors reconcile this major difference?

Answer: It is stated clearly in the Discussion section of our original version of manuscript that Cx43 is responsible for intercellular propagation of impulses and Nav1.5 accounts for intracellular conduction of excitation. The increase in Nav1.5 expression does not necessarily imply an increase in intracellular electrical conduction, simply because the membrane is depolarized in HF and such depolarization can likely render inactivation of Na⁺ channels and slow the intracellular conduction. Even if upregulation of Nav1.5 could increase intracellular conduction, the impulses would not be able to propagate to the next cardiac fibers in the absence of Cx43.

Nav1.5 levels may be increased, but to equate the level of protein expression with a functional parameter, i.e. “intracellular conduction” – is not justified. There are no data indicating functionality of the sodium channels.

Answer: It is commonly accepted that Nav1.5 accounts for intracellular conduction. We did not do any further investigations on Nav1.5 and the Na⁺ current it carries, simply because the direction of the changes of Nav1.5 is not expected to provide any rational explanations for the conduction slowing induced by CCRR downregulation or knockdown.

Non-quantitative terms like “an impressive quantity...” probably are not the best to use.

Answer: We have removed all non-quantitative terms of descriptions and statements throughout the manuscript.

The manuscript will require careful editing to improve grammar.

Answer: We have double checked our manuscript for any issues on the writing and grammar.

Reviewer #2 (expert in cardiac physiology and connexin-43)

Remarks to the Author:

This manuscript presents the intriguing possibility that lncRNA AK045950 (“CCRR^{*}”) regulates Cx43 trafficking and is reduced in heart failure, decreasing Cx43 based cell-cell coupling and increasing arrhythmogenesis. The evidence of CCRR reduction in whole hearts and of intercalated disc structural changes, reduced EF, as well as the evidence of arrhythmia induction with reduced CCRR are convincing. However the models used, mechanism of arrhythmogenesis related to Cx43 trafficking, and explanation why CCRR reduces ejection fraction as well as increasing arrhythmia, all require significantly more clarification. The data suggest that CCRR relates to structural proteins of cardiomyocytes or other cells that influence cardiomyocytes, important for heart function as well as intercalated disc maintenance, rather than primarily targeting gap junctions as the name CCRR suggests.

Answer: We thank you very much for your overall positive comments on our work.

We have provided point-by-point clarification on your concerns, as detailed below.

In respect to your comments “The data suggest that CCRR relates to structural proteins of cardiomyocytes or other cells that influence cardiomyocytes, important for heart function as well as intercalated disc maintenance, rather than primarily targeting gap junctions as the name CCRR suggests”, we are giving our explanations in this section.

First of all, we must emphasize that our study does not exclude the possibility that CCRR might be able to act on structural proteins of cardiomyocytes or other cells that are important to heart function and intercalated disc maintenance. Nonetheless, targeting gap junctions/Cx43 alone can reasonably justify for the observed damages of intercalated disc and impairment of overall cardiac function. It has long been recognized and accepted that cardiac muscles operate as a single functional syncytium; that is, they contract in unison. Impaired intercellular electromechanical coupling could likely disrupt the functional syncytium resulting in cardiac contractile dysfunction (this point is actually mentioned at the very beginning of our manuscript). Electromechanical coupling between cardiomyocytes is a basic requirement for coordinated mechanical activity in myocardium. Diminished presence of Cx43 in gap junctions is well known to cause discontinued impulse propagation or electrical uncoupling. This is because that cardiac contractile dysfunction may be accounted for, at least partly, by incoordination in contraction of individual myocytes secondary to fragmentation of the depolarizing wave front [1-3]. In addition, Cx43-associated gap coupling plays a significant role in the regulation of resting Ca²⁺ signaling in normal ventricular myocytes [4, 5], loss of which could badly hurt contractile function.

While the primary concern about the function of Cx43 has been on cardiac

excitation conduction/arrhythmogenesis, a decent number of studies have been publishing demonstrating the role of Cx43 in cardiac contractile function. Indeed, it has been generally believed that downregulation of Cx43 in human failing heart has been implicated not only in arrhythmogenesis but in contractile dysfunction as well.

For example, early in 1998, Kaprielian *et al* observed that a reduction in gap junction coupling is involved in the pathophysiology of hibernation in patients with ischaemic heart disease, possibly by unmasking and exaggerating the existing local inhomogeneities in individual cell excitability, hence disrupting wave-front propagation, slowing conduction, and leading to loss of the local coordination in myocyte contraction [1]. The authors proposed that reduction of Cx43 content in gap junctions may contribute not only to arrhythmogenesis by creating anatomic substrates of abnormal conduction but also to wall motion impairment in hibernating myocardium. In hibernating myocardium, the large Cx43 gap junctions typically found at the periphery of the intercalated disc are smaller in size, and the overall amount of immunodetectable Cx43 per intercalated disc is reduced, compared with normally perfused myocardial regions of the same heart [1]. These findings were the first indication that Cx43 gap junction remodeling contributes to impaired ventricular contraction, in addition to arrhythmia, in human ischaemic heart disease. Saffitz & Yamada supported the idea and believed that the time has come to broaden our thinking about myocardial gap junction channels beyond their electrical coupling role and to consider their possible roles in coordinating metabolic and contractile functions [2]. Apart from alterations in Cx43 gap junction organization, marked reduction in ventricular Cx43 transcript and protein levels typify the hearts of transplant patients with end-stage congestive heart failure. This Cx43 reduction occurs irrespective of whether heart failure is due to ischaemic heart disease, idiopathic dilated cardiomyopathy, or aortic stenosis [6-9]. The reduction in Cx43 is spatially heterogeneous and develops progressively during the course of disease [10]. The average reduction of 50% was observed in the failing human ventricle [8], which is consistent to our finding in the present study.

That heterogeneity in Cx43 expression is critical both to abnormal impulse propagation and contractile dysfunction is elegantly demonstrated experimentally by Gutstein *et al* in chimeric mice formed from Cx43-deficient embryonic stem cells and wild-type recipient blastocysts to model the heterogeneous pattern of gap junction expression typical of diseased myocardium [11, 12]. Such a remodeling of gap junctions has been observed in many forms of heart disease. The consequent loss of synchronous ventricular activation has been hypothesized to result in diminished cardiac performance [11, 12].

Oyamada *et al* found that the expression and localization of Cx43 and gap-junctional intercellular communication are critical for the establishment of a synchronized contraction of cultured neonatal rat cardiac myocytes [13]. The same group further found that gap junctional intercellular communication

facilitates synchronized contraction of cardiac myocytes through maintaining synchronous intracellular Ca^{2+} fluctuations [5]. An elegant study documented by Valencik et al demonstrated that conditional expression of integrin in mice causes 80% reduction in amplitude of the QRS complex, profound systolic dysfunction, decreased Cx43, loss of gap junctions, and abnormal intercalated discs. Yet, isolated left ventricular myocytes contract normally and exhibit normal Ca^{2+} transients, suggesting that intercell electromechanical coupling has been disrupted [14].

While the above studies demonstrated that loss of Cx43 can render contractile dysfunction, the following experiments show that gain-of-function of Cx43 improves cardiac function. One study found that exogenous Cx43-expressing autologous skeletal myoblasts ameliorate mechanical function and electrical activity of the rabbit heart after experimental infarction [15]. Similar observations were reported by another study: cardiomyocytes can form electromechanical junctions with some skeletal myotubes in coculture and induce their synchronous contraction via gap junctions [16]. In addition, Lakkisto *et al* observed that the heme oxygenase inducer hemin protects against cardiac dysfunction in ischemic/reperfused rat hearts a Cx43-dependent mechanism [17]. Another pharmacological study also claimed the involvement of Cx43 in the protective effects of diltiazem on cardiac function (contraction force) during hypoxic injury [18].

Despite this rational argument, we have added a brief discussion providing the alternative explanation based on the possible structural alterations for our observations (page 11, lines 5-9).

1. Kaprielian RR, Gunning M, Dupont E, Sheppard MN, Rothery SM, Underwood R, Pennell DJ, Fox K, Pepper J, Poole-Wilson PA, Severs NJ. Downregulation of immunodetectable connexin43 and decreased gap junction size in the pathogenesis of chronic hibernation in the human left ventricle. *Circulation*. 1998;97:651-60.
2. Saffitz JE, Yamada KA. Do alterations in intercellular coupling play a role in cardiac contractile dysfunction? *Circulation*. 1998;97:630-2.
3. Severs NJ, Bruce AF, Dupont E, Rothery S. Remodelling of gap junctions and connexin expression in diseased myocardium. *Cardiovasc Res*. 2008;80:9-19.
4. Li C, Meng Q, Yu X, Jing X, Xu P, Luo D. Regulatory effect of connexin 43 on basal Ca^{2+} signaling in rat ventricular myocytes. *PLoS One*. 2012;7:e36165.
5. Kimura H, Oyamada Y, Ohshika H, Mori M, Oyamada M. Reversible inhibition of gap junctional intercellular communication, synchronous contraction, and synchronism of intracellular Ca^{2+} fluctuation in cultured neonatal rat cardiac myocytes by heptanol. *Exp Cell Res*. 1995;220:348-56.
6. Kostin S, Rieger M, Dammer S, Hein S, Richter M, Klovekorn WP et al. Gap junction remodeling and altered connexin43 expression in the failing human heart. *Mol Cell Biochem* 2003;242:135–144.
7. Kostin S, Dammer S, Hein S, Klovekorn WP, Bauer EP, Schaper J. Connexin 43 expression and distribution in compensated and decompensated cardiac hypertrophy in patients with aortic stenosis. *Cardiovasc Res* 2004;62:426–436.

8. Dupont E, Matsushita T, Kaba R, Vozzi C, Coppen SR, Khan N et al. Altered connexin expression in human congestive heart failure. *J Mol Cell Cardiol* 2001;33:359–371.
9. Kitamura H, Ohnishi Y, Yoshida A, Okajima K, Azumi H, Ishida A et al. Heterogeneous loss of connexin43 protein in nonischemic dilated cardiomyopathy with ventricular tachycardia. *J Cardiovasc Electrophysiol* 2002;13:865–870.
10. Wiegerinck RF, van Veen TA, Belterman CN, Schumacher CA, Noorman M, de Bakker JM, Coronel R. Transmural dispersion of refractoriness and conduction velocity is associated with heterogeneously reduced connexin43 in a rabbit model of heart failure. *Heart Rhythm*. 2008;5:1178-85.
11. Gutstein DE, Morley GE, Vaidya D, Liu F, Chen FL, Stuhlmann H, Fishman GI. Heterogeneous expression of gap junction channels in the heart leads to conduction defects and ventricular dysfunction. *Circulation*. 2001;104:1194-9.
12. Gutstein DE, Morley GE, Fishman GI. Conditional gene targeting of connexin43: exploring the consequences of gap junction remodeling in the heart. *Cell Commun Adhes*. 2001;8:345-8.
13. Oyamada M, Kimura H, Oyamada Y, Miyamoto A, Ohshika H, Mori M. The expression, phosphorylation, and localization of connexin 43 and gap-junctional intercellular communication during the establishment of a synchronized contraction of cultured neonatal rat cardiac myocytes. *Exp Cell Res*. 1994;212:351-8.
14. Valencik ML, Zhang D, Punske B, Hu P, McDonald JA, Litwin SE. Integrin activation in the heart: a link between electrical and contractile dysfunction? *Circ Res*. 2006;99:1403-10.
15. Antanavičiūtė I, Ereminienė E, Vysockas V, Račkauskas M, Skipskis V, Rysevaitė K, Treinys R, Benetis R, Jurevičius J, Skeberdis VA. Exogenous connexin43-expressing autologous skeletal myoblasts ameliorate mechanical function and electrical activity of the rabbit heart after experimental infarction. *Int J Exp Pathol*. 2015;96:42-53.
16. Reinecke H, MacDonald GH, Hauschka SD, Murry CE. Electromechanical coupling between skeletal and cardiac muscle. Implications for infarct repair. *J Cell Biol*. 2000;149:731-40.
17. Lakkisto P, Csonka C, Fodor G, Bencsik P, Voipio-Pulkki LM, Ferdinandy P, Pulkki K. The heme oxygenase inducer hemin protects against cardiac dysfunction and ventricular fibrillation in ischaemic/reperfused rat hearts: role of connexin 43. *Scand J Clin Lab Invest*. 2009;69:209-18.
18. Matsushita S, Kurihara H, Watanabe M, Okada T, Sakai T, Amano A. Inhibition of connexin43 dephosphorylation is involved in protective effects of diltiazem on cardiac function during hypoxic injury. *Histol Histopathol*. 2011;26:315-22.

1. FISH data in mouse cardiac muscle (Figure 1b) are statistically significant but quantitatively small and, based on the representative images, can be confounded by a high background signal. The modest reductions in myocytes in the setting of large reductions in mouse and human hearts suggest a non-cardiomyocyte contribution.

Answer: We thank you for raising this issue. We do agree that the strong

background noise might have rendered significant underestimation of the reduction of CCRR in the myocardium. To tackle this problem, we have re-analyzed the images after tuning down the background. In this way, the noise was minimized and the signal/noise ratio was enhanced. After such an adjustment, the decrease in CCRR in HF is better manifested (Fig. 1c).

2. Reduced EF and disruption of the intercalated disc with rescue by CCRR are evidence of a structural consequence of CCRR rather than electrical. Cx43 gap junctions are likely the final read-out of an intact disc and cytoskeleton delivery apparatus. It is quite likely that CCRR does not directly affect Cx43 protein but rather disc structure which then can affect both Cx43 delivery and internalization.

Answer: We entirely agree with you on there is a possibility for CCRR to act on the structure of the intercalated disc leading to electrical anomalies. However, we have provided several lines of evidence for the direct effect of CCRR on Cx43 as the primary mechanism for the altered electrical activities. First, our data obtained from electron microscopy clearly show the disruption of intercalated discs; this result itself is against the view that “Cx43 gap junctions are likely the final read-out of an intact disc”. Second, in cultured and dispersed cardiomyocytes lacking the intercalated disc, similar decrease in Cx43 was consistently observed in the presence of siCCRR to silence endogenous CCRR, which could be effectively reversed by CCRR overexpression (Fig. 3d & 3e; Supplementary Figure 7). And finally, CCRR could interact directly with CIP85 (Fig. 4a), an action likely relieving the internalization of Cx43, which is clearly indicated by the increases in both membrane and cytoplasmic levels of Cx43 proteins (Fig. 3c & 3e). According to these arguments and the current absence of direct experimental evidence for the possibility that CCRR acts on the structure of the intercalated disc, we believe it makes better sense to interpret our data as the direct effect of Cx43. Another piece of evidence in support of this notion is our histological examination showing the lack of effects of CCRR on cardiac fibrosis (see the images below). Nonetheless, to avoid neglecting the possibility, we have added a statement in this regard in the revised manuscript (Page 11, lines 5-9).

Reduced EF does not necessarily imply a structural consequence of CCRR; rather it could be reasonably explained by the reduced availability of Cx43. It is well known that cardiac muscles operate as a single functional syncytium; that is, they contract in unison. Impaired electrical coupling could likely disrupt the functional syncytium resulting in contractile dysfunction. Diminished presence of Cx43 in gap junctions is well known to cause discontinued impulse propagation or electrical uncoupling. Evidence in support of this possibility has been well documented in numerous published studies. More detailed explanations have been provided at the beginning of our Responses to your concerns.

Histological assessment of cardiac fibrosis with Masson's trichrome staining of cardiac sections under various conditions. Note that HF induced significant fibrosis and neither CCRR overexpression nor CCRR knockdown altered the cardiac fibrosis.

3. CIP85 changes are modest at best and appear too low to explain significant changes in Cx43 distribution. Other key factors have been identified with dramatic effects on Cx43 internalization such as phosphorylation status of casein kinase (CK) serines as well as serine 373 and 368. It is not clear why the investigation was restricted to CIP85. It is actually not clear why, assuming a primary defect in Cx43 trafficking, the investigation assumed the trafficking affect was internalization and not forward delivery of Cx43.

Answer: From our immunostaining results, it is clear that CIP85 is highly expressed in the cell relative to Cx43 (Figure 3e & 3f; Supplementary Figures 7 & 8). Changes of small to moderate portions of CIP85 represent changes of a large number of CIP85 protein molecules, which is expected to bring about a great changes of Cx43. We therefore cannot agree with you on "CIP85 changes are modest at best and appear too low to explain significant changes in Cx43 distribution."

There could be more than one mechanisms for the actions of CCRR, just like for all other regulatory RNAs or even proteins, but we cannot test every possibility in one study. Our focus was on CIP85, and the reason for that is simply that our data point to this direction: the direction of internalization regulated by CIP85. While changes of Cx43 phosphorylation status can alter Cx43 function, they are not expected to alter Cx43 expression or distribution in the membrane. Yet, the latter is the main finding in the present study. Thus, there is not much justification for us to go with the low possibility.

It is again possible that the forward delivery process of Cx43 was also affected and our study does not try in any way to exclude such a possibility. However, we must focus on a specific direction, and most importantly, our

results on CIP85 argues for its contribution to the observed loss of Cx43 in the cytoplasmic membrane.

4. The definition of HF, reduced EF, used for the TAC mice is invalid. HF is a clinical syndrome of cardiac origin volume overload. Reduced EF (< 50%) suggests the onset of dilated cardiomyopathy but does not provide evidence that the mice have HF. Body weight, lung weight, and physical activity measures are typically used to establish HF.

Answer: We are sort of being confused by your point. According to clinical interpretations, reduction of EF means decompensation of the heart or heart failure. In certain cases, patients can have a normal EF reading but still have heart failure (called heart failure with preserved ejection fraction). Patients with reduced EF are referred to as systolic heart failure. Transverse aortic constriction (TAC) in mice is a widely and frequently used experimental model for pressure overload-induced cardiac hypertrophy and HF. TAC initially leads to compensated hypertrophy of the heart, which often is associated with a temporary enhancement of cardiac contractility. Over time, however, the response to the chronic hemodynamic overload becomes maladaptive, resulting in cardiac dilatation and heart failure. Numerous studies have shown that TAC for 4-8 weeks is sufficient for HF to develop in mice [1-17], and our experimental observations and measurements were taken 8 weeks after TAC.

In our study EF was substantially reduced, which was accompanied by systolic impairments, and this is a strong indication of HF, as opposed to compensated heart with normal EF. Moreover, we did not solely use EF as an index for HF, but have also presented other parameters of cardiac function (Supplementary Tables 1-4) in support of the case. For example, the decreases in LVPWs (thickness of systolic left ventricular posterior wall), LVPWd (thickness of diastolic left ventricular posterior wall), and LVPWs (thickness of systolic left ventricular posterior wall) are most reasonably interpreted as decompensated myocardium or heart failure. Furthermore, it is known that pressure overload induced congestive HF is manifested by reduction in fractional shortening (FS)]. Our data clearly demonstrated the remarkable reduction in FS (Supplementary Tables 1-4).

Despite these above arguments, we have now also provided the data on body weight/body weight and lung weight/body weight ratios as additional evidence for HF (Supplementary Figure 11). As expected, TAC resulted in greater heart weight- and lung weight-to-body weight ratios in HF mice and the mice treated with Lv-siCCRR.

1. Yu J, et al. Salvianolic Acid B Alleviates Heart Failure by Inactivating ERK1/2/GATA4 Signaling Pathway after Pressure Overload in Mice. *PLoS One*. 2016;11(11):e0166560.
2. Shirakabe A, et al. Drp1-Dependent Mitochondrial Autophagy Plays a Protective Role Against Pressure Overload-Induced Mitochondrial Dysfunction and Heart Failure. *Circulation*. 2016;133(13):1249-63.

3. Laroumanie F, et al. CD4+ T cells promote the transition from hypertrophy to heart failure during chronic pressure overload. *Circulation*. 2014;129(21):2111-24.
4. Dai DF, et al. Mitochondrial proteome remodelling in pressure overload-induced heart failure: the role of mitochondrial oxidative stress. *Cardiovasc Res*. 2012;93(1):79-88.
5. Okada K, et al. Prolonged endoplasmic reticulum stress in hypertrophic and failing heart after aortic constriction: possible contribution of endoplasmic reticulum stress to cardiac myocyte apoptosis. *Circulation*. 2004;110(6):705-12.
6. Liao Y, et al. Activation of adenosine A1 receptor attenuates cardiac hypertrophy and prevents heart failure in murine left ventricular pressure-overload model. *Circ Res*. 2003;93(8):759-66.
7. Liao Y, et al. Deficiency of type 1 cannabinoid receptors worsens acute heart failure induced by pressure overload in mice. *Eur Heart J*. 2012;33(24):3124-33.
8. Riehle C, et al. PGC-1 β deficiency accelerates the transition to heart failure in pressure overload hypertrophy. *Circ Res*. 2011;109(7):783-93.
9. Willis MS, et al. Cardiac muscle ring finger-1 increases susceptibility to heart failure in vivo. *Circ Res*. 2009;105(1):80-8.
10. Ling H, et al. Requirement for Ca²⁺/calmodulin-dependent kinase II in the transition from pressure overload-induced cardiac hypertrophy to heart failure in mice. *J Clin Invest*. 2009;119(5):1230-40.
11. Grote-Wessels S, et al. Inhibition of protein phosphatase 1 by inhibitor-2 exacerbates progression of cardiac failure in a model with pressure overload. *Cardiovasc Res*. 2008 Aug 1;79(3):464-71.
12. Loyer X, et al. Cardiomyocyte overexpression of neuronal nitric oxide synthase delays transition toward heart failure in response to pressure overload by preserving calcium cycling. *Circulation*. 2008;117(25):3187-98.
13. Liao Y, et al. Control of plasma glucose with alpha-glucosidase inhibitor attenuates oxidative stress and slows the progression of heart failure in mice. *Cardiovasc Res*. 2006;70(1):107-16.
14. Patel B, Ismahil MA, Hamid T, Bansal SS, Prabhu SD. Mononuclear Phagocytes Are Dispensable for Cardiac Remodeling in Established Pressure-Overload Heart Failure. *PLoS One*. 2017;12:e0170781.
15. Kaimoto S, Hoshino A, Ariyoshi M, Okawa Y1, Tateishi S, Ono K, Uchihashi M, Fukai K, Iwai-Kanai E2, Matoba S. Activation of PPAR α in the early stage of heart failure maintained myocardial function and energetics in pressure overload heart failure. *Am J Physiol Heart Circ Physiol*. 2016 Dec 23:ajpheart.00553.2016.
16. Hampton C, Rosa R, Campbell B, Kennan R, Gichuru L, Ping X, Shen X, Small K, Madwed J, Lynch JJ. Early echocardiographic predictors of outcomes in the mouse transverse aortic constriction heart failure model. *J Pharmacol Toxicol Methods*. 2016;84:93-101.
17. Kato T, et al. Correction of impaired calmodulin binding to RyR2 as a novel therapy for lethal arrhythmia in the pressure-overloaded heart failure. *Heart Rhythm*. 2017;14(1):120-127.

5. The human samples of HF have considerable variability. Several of the samples of preserved ejection fraction while others have reduced ejection fraction. Two of the patients had mitral stenosis which could result in heart failure yet with an unaffected ventricle. A uniform reduced of CCRR in these samples argues against HF induced reductions and may be due either to other forms of cardiac disease or, quite possibly, an artifact of obtained samples from deceased individuals.

Answer: It is not unexpected that human samples are more variable compared to animals of pure breed; even the seemingly healthy subjects can have considerable variations. By comparing with numerous published studies, it becomes clear that the variation of our human data is reasonable and smaller than many other studies. To validate this argument, we have provided a scatterplot of the all individual human data in Figure 1b (right panel), which demonstrates better-than-acceptable variations of our data. Also an important point is that the patients included in our study have a common diagnosis (they were all diagnosed with HF, but not as perceived by your guess “mitral stenosis which could result in heart failure yet with an unaffected ventricle”) and a common change of CCRR (they all had a decreased CCRR level). The most rational and straightforward interpretation of these data is therefore that CCRR is downregulated in HF patients. One can always seek more complicated explanations for the observations, but that does not help clarify the issue. We could not agree with you on the point that a uniformly reduced CCRR in these samples argues against HF induced reductions. Conversely, our data indicate that regardless of how or by what kinds of etiology the patients developed HF, they all have significantly higher probability of CCRR downregulation.

Reviewer #3 (expert in cardiac physiology and lncRNA)

Remarks to the Author:

In this report, Sun et al, reported the functional characterization of an lncRNA, termed CCRR in cardiac conduction regulation. The expression of CCRR was measured in heart failure tissue. Loss- and gain-of function studies in mouse hearts and cultured myocytes were conducted to evaluate the impact of CCRR. In addition to slowing electric conduction, arrhythmogenesis and ultrastructure, changes in cardiac function and gene expression were also observed. Furthermore, CCRR impact on Cx43/CIP85 expression and distribution, as well as its interaction with Cx43/CIP85 complex was characterized. Finally, NFAT3 mediated regulation of CCRR down-regulation was demonstrated.

The concept of an lncRNA mediated modulation of cardiac conduction via direct interaction with Cx43/CIP85 complex is novel and potentially important based on in vivo gain/loss-of-function analysis. However, the current report appears to be quite preliminary in developing this hypothesis and in revealing the underlying mechanisms. The validity of the central conclusion should also be better demonstrated.

Answer: We are grateful to you for your very positive view and comments on our study. We have provided below the point-by-point explanations for the specific concerns you raised.

1). CCRR connection with abnormal EP in heart failure. Authors indicated that informatic analysis based on co-expression network of lncRNAs and coding RNAs in failing heart led to the initial hypothesis that CCRR was functionally linked to cardiac conduction. However, no specific details were provided to demonstrate how such connection was established and how significant of this connection from informatics analysis. In fact, both data from gain and loss of function studies appeared to indicate CCRR had a broad impact on cardiac pathology in addition to conduction defects, such as contractile dysfunction.

Answer: The detailed information is provided in our published study, as mentioned in the main text (page 5, lines 5-7): “Bioinformatics analyses (GO analysis, pathway analysis, and lncRNAs-mRNAs co-expression network analysis) enabled us to identify a subset of lncRNAs of high profiles linking to HF¹⁶”.

In respect to your comments “In fact, both data from gain and loss of function studies appeared to indicate CCRR had a broad impact on cardiac pathology in addition to conduction defects, such as contractile dysfunction”, we are giving our explanations below. First of all, we must emphasize that our study does not exclude the possibility that CCRR might be able to act on structural proteins of cardiomyocytes or other cells that are important to heart function and intercalated disc maintenance. Nonetheless, targeting gap junctions/Cx43 alone provides reasonable explanations for the observed

damages of intercalated disc and impairment of overall cardiac function. It has long been recognized and accepted that cardiac muscles operate as a single functional syncytium; that is, they contract in unison. Impaired intercellular electromechanical coupling could likely disrupt the functional syncytium resulting in contractile dysfunction (this point is actually mentioned at the very beginning of the original version of our manuscript). Electromechanical coupling between cardiomyocytes is a basic requirement for coordinated mechanical activity in myocardium. Diminished presence of Cx43 in gap junctions is well known to cause discontinued impulse propagation or electrical uncoupling. This is because that cardiac contractile dysfunction may be accounted for, at least partly, by incoordination in contraction of individual myocytes secondary to fragmentation of the depolarizing wave front [1-3]. In addition, Cx43-associated gap coupling plays a significant role in the regulation of resting Ca^{2+} signaling in normal ventricular myocytes [4, 5], loss of which can severely hurt contractile function.

While the primary concern about the function of Cx43 has been on cardiac excitation conduction/arrhythmogenesis, a decent number of studies have been publishing demonstrating the role of Cx43 in cardiac contractile function. Indeed, it has been generally believed that downregulation of Cx43 in human failing heart has been implicated not only in arrhythmogenesis but in contractile dysfunction as well.

For example, early in 1998, Kaprielian *et al* observed that a reduction in gap junction coupling is involved in the pathophysiology of hibernation in patients with ischaemic heart disease, possibly by unmasking and exaggerating the existing local inhomogeneities in individual cell excitability, hence disrupting wave-front propagation, slowing conduction, and leading to loss of the local coordination in myocyte contraction [1]. The authors proposed that reduction of Cx43 content in gap junctions may contribute not only to arrhythmogenesis by creating anatomic substrates of abnormal conduction but also to wall motion impairment in hibernating myocardium. In hibernating myocardium, the large Cx43 gap junctions typically found at the periphery of the intercalated disc are smaller in size, and the overall amount of immunodetectable Cx43 per intercalated disc is reduced, compared with normally perfused myocardial regions of the same heart [1]. These findings were the first indication that Cx43 gap junction remodeling contributes to impaired ventricular contraction, in addition to arrhythmia, in human ischaemic heart disease. Saffitz & Yamada supported the idea and believed that the time has come to broaden our thinking about myocardial gap junction channels beyond their electrical coupling role and to consider their possible roles in coordinating metabolic and contractile functions [2]. Apart from alterations in Cx43 gap junction organization, marked reduction in ventricular Cx43 transcript and protein levels typify the hearts of transplant patients with end-stage congestive heart failure. This Cx43 reduction occurs irrespective of whether heart failure is due to ischaemic heart disease, idiopathic dilated cardiomyopathy, or aortic stenosis [6-9]. The reduction in

Cx43 is spatially heterogeneous and develops progressively during the course of disease [10]. The average reduction of 50% was observed in the failing human ventricle [8], which is consistent to our finding in the present study.

That heterogeneity in Cx43 expression is critical both to abnormal impulse propagation and contractile dysfunction is elegantly demonstrated experimentally by Gutstein *et al* in chimeric mice formed from Cx43-deficient embryonic stem cells and wild-type recipient blastocysts to model the heterogeneous pattern of gap junction expression typical of diseased myocardium [11, 12]. Such a remodeling of gap junctions has been observed in many forms of heart disease. The consequent loss of synchronous ventricular activation has been hypothesized to result in diminished cardiac performance [11, 12].

Oyamada *et al* found that the expression, phosphorylation, and localization of Cx43 and gap-junctional intercellular communication are critical for the establishment of a synchronized contraction of cultured neonatal rat cardiac myocytes [13]. The same group further found that gap junctional intercellular communication facilitates synchronized contraction of cardiac myocytes through maintaining synchronous intracellular Ca²⁺ fluctuations [5]. An elegant study documented by Valencik *et al* demonstrated that conditional expression of integrin in mice causes 80% reduction in amplitude of the QRS complex, profound systolic dysfunction, decreased Cx43, loss of gap junctions, and abnormal intercalated discs. Yet, isolated left ventricular myocytes contract normally and exhibit normal Ca²⁺ transients, suggesting that intercell electromechanical coupling has been disrupted [14].

While the above studies demonstrated that loss of Cx43 can render contractile dysfunction, the following experiments show that gain-of-function of Cx43 improves cardiac function. One study found that exogenous Cx43-expressing autologous skeletal myoblasts ameliorate mechanical function and electrical activity of the rabbit heart after experimental infarction [15]. Similar observations were reported by another study: cardiomyocytes can form electromechanical junctions with some skeletal myotubes in coculture and induce their synchronous contraction via gap junctions [16]. In addition, Lakkisto *et al* observed that the heme oxygenase inducer hemin protects against cardiac dysfunction in ischemic/reperfused rat hearts a Cx43-dependent mechanism [17]. Another pharmacological study also claimed the involvement of Cx43 in the protective effects of diltiazem on cardiac function (contraction force) during hypoxic injury [18].

Despite this rational argument, we have added a brief discussion providing the alternative explanation based on the possible structural alterations for our observations (page 11, lines 5-9).

1. Kaprielian RR, Gunning M, Dupont E, Sheppard MN, Rothery SM, Underwood R, Pennell DJ, Fox K, Pepper J, Poole-Wilson PA, Severs NJ. Downregulation of immunodetectable connexin43 and decreased gap junction size in the pathogenesis of chronic hibernation in the human left ventricle. *Circulation*. 1998;97:651-60.

2. Saffitz JE, Yamada KA. Do alterations in intercellular coupling play a role in cardiac contractile dysfunction? *Circulation*. 1998;97:630-2.
3. Severs NJ, Bruce AF, Dupont E, Rothery S. Remodelling of gap junctions and connexin expression in diseased myocardium. *Cardiovasc Res*. 2008;80:9-19.
4. Li C, Meng Q, Yu X, Jing X, Xu P, Luo D. Regulatory effect of connexin 43 on basal Ca²⁺ signaling in rat ventricular myocytes. *PLoS One*. 2012;7:e36165.
5. Kimura H, Oyamada Y, Ohshika H, Mori M, Oyamada M. Reversible inhibition of gap junctional intercellular communication, synchronous contraction, and synchronism of intracellular Ca²⁺ fluctuation in cultured neonatal rat cardiac myocytes by heptanol. *Exp Cell Res*. 1995;220:348-56.
6. Kostin S, Rieger M, Dammer S, Hein S, Richter M, Klovekorn WP et al. Gap junction remodeling and altered connexin43 expression in the failing human heart. *Mol Cell Biochem* 2003;242:135–144.
7. Kostin S, Dammer S, Hein S, Klovekorn WP, Bauer EP, Schaper J. Connexin 43 expression and distribution in compensated and decompensated cardiac hypertrophy in patients with aortic stenosis. *Cardiovasc Res* 2004;62:426–436.
8. Dupont E, Matsushita T, Kaba R, Vozzi C, Coppens SR, Khan N et al. Altered connexin expression in human congestive heart failure. *J Mol Cell Cardiol* 2001;33:359–371.
9. Kitamura H, Ohnishi Y, Yoshida A, Okajima K, Azumi H, Ishida A et al. Heterogeneous loss of connexin43 protein in nonischemic dilated cardiomyopathy with ventricular tachycardia. *J Cardiovasc Electrophysiol* 2002;13:865–870.
10. Wiegerinck RF, van Veen TA, Belterman CN, Schumacher CA, Noorman M, de Bakker JM, Coronel R. Transmural dispersion of refractoriness and conduction velocity is associated with heterogeneously reduced connexin43 in a rabbit model of heart failure. *Heart Rhythm*. 2008;5:1178-85.
11. Gutstein DE, Morley GE, Vaidya D, Liu F, Chen FL, Stuhlmann H, Fishman GI. Heterogeneous expression of gap junction channels in the heart leads to conduction defects and ventricular dysfunction. *Circulation*. 2001;104:1194-9.
12. Gutstein DE, Morley GE, Fishman GI. Conditional gene targeting of connexin43: exploring the consequences of gap junction remodeling in the heart. *Cell Commun Adhes*. 2001;8:345-8.
13. Oyamada M, Kimura H, Oyamada Y, Miyamoto A, Ohshika H, Mori M. The expression, phosphorylation, and localization of connexin 43 and gap-junctional intercellular communication during the establishment of a synchronized contraction of cultured neonatal rat cardiac myocytes. *Exp Cell Res*. 1994;212:351-8.
14. Valencik ML, Zhang D, Punske B, Hu P, McDonald JA, Litwin SE. Integrin activation in the heart: a link between electrical and contractile dysfunction? *Circ Res*. 2006;99:1403-10.
15. Antanavičiūtė I, Ereminienė E, Vysockas V, Račkauskas M, Skipskis V, Rysevaitė K, Treinys R, Benetis R, Jurevičius J, Skeberdis VA. Exogenous connexin43-expressing autologous skeletal myoblasts ameliorate mechanical function and electrical activity of the rabbit heart after experimental infarction. *Int J Exp Pathol*. 2015;96:42-53.

16. Reinecke H, MacDonald GH, Hauschka SD, Murry CE. Electromechanical coupling between skeletal and cardiac muscle. Implications for infarct repair. *J Cell Biol.* 2000;149:731-40.
17. Lakkisto P, Csonka C, Fodor G, Bencsik P, Voipio-Pulkki LM, Ferdinandy P, Pulkki K. The heme oxygenase inducer hemin protects against cardiac dysfunction and ventricular fibrillation in ischaemic/reperfused rat hearts: role of connexin 43. *Scand J Clin Lab Invest.* 2009;69:209-18.
18. Matsushita S, Kurihara H, Watanabe M, Okada T, Sakai T, Amano A. Inhibition of connexin43 dephosphorylation is involved in protective effects of diltiazem on cardiac function during hypoxic injury. *Histol Histopathol.* 2011;26:315-22.

2). CCRR expression: Authors used both RT-PCR and FISH to detect CCRR expression. For lncRNA, it is important to validate the expression of full length transcript or other potential isoforms by northern blots. In addition, the FISH analysis in Figure 1 was not adequate in quality and amplifications, and lacked positive and negative controls. It is not clear the cellular structure associated with the enriched staining. But the pattern does not fit cell-cell junction or correlated with localization of Cx43/CIP85. This is a major concern.

Answer: We have performed additional experiments using Northern blot analysis. The results are in perfect agreement with our original conclusion about the downregulation of CCRR in HF. The data is provided below (please see the image below). We do not include this data in our manuscript because it does not provide any new information beyond our data obtained from qPCR and FISH experiments, but will make our manuscript unnecessarily lengthy.

For additional FISH experiments, we added a negative control probe for the specificity of the CCRR probe used in our study (Figure 1c).

Representative Northern blotting bands showing the downregulation of CCRR in the myocardium of HF mice induced by TAC, relative to the sham-operated control mice, *P<0.05, HF vs. Ctl; n=4. CCRR: CCRR-specific probe; Actin: Actin-specific probe as a internal control.

3). Function of CCRR: authors presented evidence to support the effect of over-expression or siRNA KD on mouse heart function and cardiac conduction. As shown in Supplemental Figure 2, the KD effect was very modest comparing to the

observed reduction in HF samples. It is also not clear about the duration of CCRR over-expression throughout the development of HF (it is not clear at what time points during the experiment the data in Supplemental Figure 2 was obtained). In addition, the KO or the overexpression by lentiviral vectors were performed without targeted cell-type specificity. Therefore, it is not clear in these experiments, the observed changes in cardiac phenotype were entirely due to their effect in cardiomyocytes. In addition to inter-cellular coupling, interstitial fibrosis can be a major contributor to conduction defects and arrhythmia. Authors observed significant changes in cardiac function in addition to cardiac conduction or rhythm. These changes may be related to each other. However, without more detailed temporal analysis and targeted intervention, it is not clear if cardiac conduction defect is a direct and specific outcome of CCRR expression or indirectly through contractile dysfunction.

Answer:

We conducted additional experiments verifying the efficacy of Lv-siCCRR in knocking down endogenous CCRR. With increased number of experiments, the variations were minimized and the knockdown was better manifested. The new data are incorporated into Supplementary Figure 5b & 5d. It should also be noted that our measurements were made 9 weeks post administration of the construct, and the effect of Lv-siCCRR is expected to be reduced over such a long time-span.

The viral constructs were administered into mouse heart by intra-cavity injection (injected into left ventricular chamber) one week before TAC surgery. This is stated in METHODS (page 16, lines 9-10).

All our measurements were made eight weeks following TAC surgery (This is stated in our manuscript; page 13, lines 4-6).

The fact that CCRR can reproduce its effects on Cx43 and CIP85 in cultured cardiomyocytes (Supplementary Figures 6-8) argues for its direct effects on cardiomyocytes in the in vivo setting, though the possible effects on other cell types cannot be ruled out.

We used conventional approach for virus infection and such an approach cannot rule out the possibility that the effects of CCRR on non-cardiomyocytes contribute to the observed changes in cardiac phenotype. Nonetheless, the key issue is that our data clearly demonstrated the direct effects of CCRR on Cx43; and we do not see any reason to take the non-straightforward data interpretations instead and to complicate the issue. For any piece of study (published or undergoing), multiple possibilities always and inevitably exist, but one can just focus on the one that gives the best explanation.

Our histological examinations failed to support the action of CCRR on cardiac fibrosis (please see below for the results).

Our explanations for the possibility of CCRR acting on contractile dysfunction have been provided above for your question #1.

Histological assessment of cardiac fibrosis with Masson's trichrome staining of cardiac sections under various conditions. Note that HF induced significant fibrosis and neither CCRR overexpression nor CCRR knockdown altered the cardiac fibrosis.

- A. The specific impact of CCRR expression on cardiomyocyte inter-cellular conduction should be demonstrated directly in cultured myocytes based on the effect of its gain/loss of CCRR expression on Cx43/CIP85 expression or distribution and cell-cell junction (by dye transfer). This would be also a suitable system to interrogate the mechanisms.

Answer: As per your suggestion, we have conducted additional experiments with immunocyto staining, Western blot, and scrape loading and dye transfer techniques in cultured adult human ventricular cardiomyocyte AC16 cells. Our results provided an additional line of evidence in support of our original view that CCRR acts directly on cell-cell junction. The new data are shown in Supplementary Figures 4 & 6-9. In particular, the data illustrated in Supplementary Figure 9 confirmed the role of CCRR in regulating intercellular communications (presumably through gap junction channels) using a diffusible dye (Lucifer yellow): Lv-CCRR pronouncedly promoted, whereas Lv-siCCRR nearly completely blocked, the diffusion of Lucifer yellow.

4). Mechanism of CCRR function: Authors provided intriguing evidence that CCRR interacted with CIP85 directly and regulated Cx43 distribution at gap junction via CIP85/Cx43 complex. However, there are several significant issues with the supporting data.

- A. The data shown in Figure 3C/D is puzzling regarding the intracellular distribution. Cx43 is synthesized and inserted into the ER membrane, and then transported to cytoplasmic membrane followed by recycling back into lysosome for degradation or reinsertion. In heart, majority of Cx43 protein signaling should be detected on membrane. Authors should provide more details about the conditions used for

fractionation (detergent content) and additional data to support the quality of the fractionation method. The same concern applies to CIP85, except in this case, the immunostaining shown in Figure 3e did not support discrete pattern of CIP85 distribution. The colocalization with Cx43 was clearly over-interpreted based on the overly saturated CIP85 signal covering the entire myocyte.

Answer: Cx43, like any other membrane proteins, is synthesized in the endoplasmic reticulum and transported to the cytoplasmic membrane. Clearly, it exist in both cytoplasm and membrane. It is expected to see Cx43 signals in both cytoplasmic and membrane fractions with Western blotting using extracted protein samples. For immunocytochemistry, however, whether Cx43 can be visualized depends upon the antibody used for detection. If the antibody is directed to target the extracellular portion of Cx43, then cytoplasmic Cx43 is normally unable to be seen. In our study, the antibody is generated from the antigenic epitope of the extracellular region of Cx43 protein. Hence, it can recognize only the membrane Cx43 for immunocytostaining but both cytoplasmic and membrane Cx43 for immunoblotting.

As for CIP85, this protein is not supposed to have discrete pattern of distribution, but is ubiquitously expressed in the cytoplasm [1-4]. It is literally located within the cytoplasm, but when binding to Cx43 on the membrane, it can appear as membrane localization.

1. Cochrane K, Su V, Lau AF. The connexin43-interacting protein, CIP85, mediates the internalization of connexin43 from the plasma membrane. *Cell Commun Adhes.* 2013;20(3-4):53-66.
2. Cochrane K, Berestecky JM, Kitamura C, Lau AF. Monoclonal antibodies against the connexin43-interacting protein CIP85. *Hybridoma (Larchmt).* 2009;28(5):355-61.
3. Giepmans BN. Role of connexin43-interacting proteins at gap junctions. *Adv Cardiol.* 2006;42:41-56.
4. Lan Z, et al. Novel rab GAP-like protein, CIP85, interacts with connexin43 and induces its degradation. *Biochemistry.* 2005;44(7):2385-96.

B. Authors stated that CCRR regulates Cx43 via direct interaction with CIP85. However, the pull-down recovery with CIP85 for CCRR was very low (comparing with the total input), and there is no good control to support the specificity other than IgG. At least, authors should try to use another non-specific lncRNA as a negative control. Previous report from Lau's group indicate that CIP85 expression promotes Cx43 degradation. However, there is no data from current study to support that CIP85 interaction with CCRR had a significant impact on its binding or regulation to Cx43 distribution/cycling. Data shown in Figure 4a and b should be tested under control, HF and with CCRR manipulations. The overall hypothesis that CCRR down-regulation led to enhanced CIP85 interaction with Cx43 and its internalization and degradation lacks clear evidence and remains speculative at this time.

Answer: It should be noted here that Input is used as a control for the presence of the molecule of interest, but not as a control for quantitative comparison. The relative quantities depend on the amount of the Input, and it is

not unexpected to see stronger signals from an Input (as a matter of fact, Input normally and usually gives the strongest signal). Moreover, we have carried out additional pull-down experiments with addition of another lncRNA-ZFAS1 (Zinc finger antisense 1) and Antisense of CCRR with two as negative controls and 3' untranslated-region androgen receptor (AR) RNA as a positive control, as you suggested.

As to your suggestion "Data shown in Figure 4a and b should be tested under control, HF and with CCRR manipulations," we are afraid that the experiments could not be done. To the best of our knowledge, RIP is not a method for quantitative comparison of a RNA:protein binding with varying treatments, but is merely a way for experimental identification of RNA:protein complex [1-6]. More specifically, it is a technique of detecting the association of individual proteins with specific RNA molecules in vivo [1-6]. It can be used to investigate lncRNA-protein interaction and identify lncRNAs that bind to a protein of interest. Moreover, because of the multiple steps involved in the procedures (see the illustration below), accurate comparisons on the abundance among various treatment groups are not realistic and could not likely provide unambiguous data interpretation.

In regard with the impact of CIP85 interaction with CCRR on its regulation to Cx43 distribution/cycling, we have provided at least three lines of supporting evidence, in addition to the data shown in Figure 4a & 4b. First, our Western blot data in Fig. 3c & 3d indicate the ability of CCRR to alter the amount of CIP85 and Cx43 in the membrane fractions. CIP85 is normally localized in the cytoplasm and is associated to the membrane only when binding to Cx43. The changes of the amount of membrane-associated CIP85 by CCRR is an indication of the impact of CIP85 interaction with CCRR on Cx43 distribution/cycling. Second, such an impact is also demonstrated in Supplementary Figure 7 where Lv-siCCRR mitigated, whereas Lv-CCRR enhanced, the co-localization of CIP85 and Cx43 in the membrane. Moreover, knockdown of CIP85 also abrogated the CIP85-Cx43 co-localization in the membrane (Supplementary Figure 8).

With all these lines of supporting evidence, we are confident to hold our original view about the mechanism of CCRR action, despite that we do not intend to exclude the contributions of any other possible mechanisms.

1. McHugh CA, Russell P, Guttman M. Methods for comprehensive experimental identification of RNA-protein interactions. *Genome Biology* 2014;15:203.
2. Meier J, et al. Genome-wide identification of translationally inhibited and degraded miR-155 targets using RNA-interacting protein-IP. *RNA Biol.* 2013;10(6):1018-29.
3. Keene JD, Komisarow JM, Friedersdorf MB. RIP-Chip: the isolation and identification of mRNAs, microRNAs and protein components of ribonucleoprotein complexes from cell extracts. *Nature Protocols* 2006;1:302-307.
4. Peritz T. Immunoprecipitation of mRNA-protein complexes. *Nature Protocols* 2006;1:577-580.

5. Selth LA, Close P, Svejstrup JQ. Studying RNA–protein interactions in vivo by rna immunoprecipitation. In *Epigenetics Protocols: Second Edition*, Edited by Trygve O. Tollefsbol. *Methods in Molecular Biology* 2011;Vol 791:pp253-264. Springer Science+Business Media, LLC.
6. Zhang Y. Characterization of Long Noncoding RNA-Associated Proteins by RNA-Immunoprecipitation. *Methods Mol Biol.* 2016;1402:19-26.

Fig. 1. Flowchart of RNA immunoprecipitation protocol. (Reproduced from (10) with permission from Cold Spring Harbor Press).

(From: Selth LA, Close P, Svejstrup JQ. Studying RNA–protein interactions in vivo by rna immunoprecipitation. In *Epigenetics Protocols: Second Edition*, Edited by Trygve O. Tollefsbol. *Methods in Molecular Biology* 2011;Vol 791:pp253-264. Springer Science+Business Media, LLC.)

5). NFAT mediated regulation of CCRR down-regulation. This is an interesting observation as NFAT is usually viewed as a transcription activator but the current data only demonstrated NFAT binding to CCRR promoter regions and its expression is correlated with CCRR expression. This is not a strong evidence to support NFAT is an immediate upstream regulator of CCRR expression. Nevertheless, this is not a critical issue for this report and authors should remove it to enhance the focus of this study, i.e. the functional relevance of CCRR in cardiac pathology, particularly, in arrhythmia.

Answer: To the best of our knowledge, NFAT can act both as a transcription activator and repressor, depending on different genes. As a matter of fact, NFAT often acts as a negative regulator (repressor) of transcription of genes (just to name a few references from the literature below). Nonetheless, we

agree with you that NFAT data is not a critical issue, and thus decided to remove these data from our manuscript as suggested.

1. Nguyen T, Lindner R, Tedeschi A, Forsberg K, Green A, Wuttke A, Gaub P, Di Giovanni S. NFAT-3 is a transcriptional repressor of the growth-associated protein 43 during neuronal maturation. *J Biol Chem*. 2009;284:18816-23.
2. Luo X, Pan Z, Shan H, Xiao J, Sun X, Wang N, Lin H, Xiao L, Maguy A, Qi XY, Li Y, Gao X, Dong D, Zhang Y, Bai Y, Ai J, Sun L, Lu H, Luo XY, Wang Z, Lu Y, Yang B, Nattel S. MicroRNA-26 governs profibrillatory inward-rectifier potassium current changes in atrial fibrillation. *J Clin Invest*. 2013;123:1939-51.
3. Berglund LM, Kotova O, Osmark P, Grufman H, Xing C, Lydrup ML, Goncalves I, Autieri MV, Gomez MF. NFAT regulates the expression of AIF-1 and IRT-1: yin and yang splice variants of neointima formation and atherosclerosis. *Cardiovasc Res*. 2012;93:414-23.
4. Rana ZA, Gundersen K, Buonanno A. Activity-dependent repression of muscle genes by NFAT. *Proc Natl Acad Sci U S A*. 2008;105:5921-6.
5. Carvalho LD, Teixeira LK, Carrossini N, Caldeira AT, Ansel KM, Rao A, Viola JP. The NFAT1 transcription factor is a repressor of cyclin A2 gene expression. *Cell Cycle*. 2007;6:1789-95.
6. Kim HB, Kong M, Kim TM, Suh YH, Kim WH, Lim JH, Song JH, Jung MH. NFATc4 and ATF3 negatively regulate adiponectin gene expression in 3T3-L1 adipocytes. *Diabetes*. 2006;55:1342-52.
7. Baumgart S, Glesel E, Singh G, Chen NM, Reutlinger K, Zhang J, Billadeau DD, Fernandez-Zapico ME, Gress TM, Singh SK, Ellenrieder V. Restricted heterochromatin formation links NFATc2 repressor activity with growth promotion in pancreatic cancer. *Gastroenterology*. 2012;142:388-98.e1-7.
8. Kosiorek M, Zylinska L, Zablocki K, Pikula S. Calcineurin/NFAT signaling represses genes Vamp1 and Vamp2 via PMCA-dependent mechanism during dopamine secretion by Pheochromocytoma cells. *PLoS One*. 2014;9:e92176.

Reviewers' comments:

Reviewer #1 (Remarks to the Author):

The authors have addressed many of my previous comments. There are still a number of grammatical changes and a few technical concerns, listed below.

page 4

The entirety of cardiac muscle does not truly contract in unison. There is apex to base excitation and contraction.

"This property is gifted" is an awkward term. Please consider changing.

"Individual cardiac muscle fibers" I would call them cells, not fibers. That terminology seems more appropriate for skeletal muscle or perhaps Purkinje fibers, but not working myocardium.

"come into the scene" is an awkward term.

page 5

"manifested with" - perhaps replace with characterized by

"showed that the excitation conduction" - just simplify to showed that conduction velocity...

page 7 and elsewhere

"cytoplasmic membrane" is a confusing term. What are you actually referring to?

page 8

"pronouncedly promoted" This needs to be changed - perhaps to significantly increased

page 9

Its curious that there is a inverse relationship between CCRR and Nav1.5. Please comment further on this observation.

Figure 2.

Please clarify how the optical mapping was performed. Was CV calculated during pacing or sinus rhythm? If pacing, where was the heart paced? It says conduction from septal apex- base, however the map seems to show activation from base toward apex?? CVs are not usually calculated as described in your methods section. In addition, perhaps using a color LUT without a large grey section in the middle would improve the images.

In panel B, the sham control has an incidence of 2/13 with VT. In panel D, the sham control has an incidence of 7/20. This is quite a bit of variation for what should be two comparable control groups. Please explain.

Reviewer #2 (Remarks to the Author):

The authors' detailed response and additional studies are appreciated. No further concerns.

Reviewer #3 (Remarks to the Author):

Towards the original submission, all reviewers agreed that the findings regarding the functional impact of CCRR on cardiac conduction defect, arrhythmogenesis and overall pathological manifestation are interesting, novel and potentially important. The reviewers have also all

identified similar issues regarding the mechanistic aspects of this report, namely, the validity of CCRR mediated regulation of Cx43 degradation via CIP85 interaction. In this revision, authors provided additional evidence and clarifications to further enforce the notion that CCRR is an important lncRNA to maintain cardiac conduction and contractile function. However, the revision did not provide adequate new data to address some of the previous concerns towards the mechanistic studies and their interpretations.

1). The central hypothesis of this report (regarding mechanism) is that CCRR binds to CIP85, and blocks its negative regulation of Cx43 through endocytosis. However, other than weak binding between CCRR and CIP85, and very modest effect of CCRR expression on CIP85 distribution (cytosol vs. membrane), there are no definitive molecular evidence to support this molecular model. As authors agreed, this may be one of many possible mechanisms, but the text and conclusion leave the impression that the current model is the answer. Authors should either provide clear evidence that CCRR affects Cx43 degradation, and this effect is mediated through CIP85 and requires CIP85/CCRR interaction, or significantly revise the interpretation and conclusion.

2). The following experiments are suggested examples to support the working hypothesis:

A. CCRR regulates CIP85 intracellular distribution through direct binding: Authors have identified potential interaction between the two via RNA-Protein Interaction Prediction (RPISeq) database and validated by experiments. However, there was no data to support the functional importance of such interaction. Authors should identify binding motifs on CCRR for CIP85 interaction and demonstrate the functional role of this motif in Cx43 regulation.

B. CCRR manipulation only yielded very modest changes in CIP85 expression. If authors argue about the importance of such modest changes in Cx43 expression, they should demonstrate if CCRR changes the interaction between Cx43 and CIP85. The prediction is CCRR blocks CIP85 interaction with Cx43 at functional level more so than at its expression.

C. CIP85 mediated Cx43 expression, distribution and function: the current data provided only correlative relationship without clear indication on the directionality of the effect (forward insertion or degradation) of Cx43. This can be tested at least in cultured cells where the dependence of Cx43 downregulation on lysosomal activity can be directly demonstrated under CCRR manipulation.

3). There are still numerous places where non-quantitative adjectives are used such as "enormous", "pronouncedly" to describe the quantitative changes.

Responses to Reviewers' Comments

Reviewer #1 (Remarks to the Author):

The authors have addressed many of my previous comments. There are still a number of grammatical changes and a few technical concerns, listed below.

Responses: We thank you for your positive view of our work. We have modified our manuscript to meet the grammatical satisfaction and have also clarified the technical points, as detailed below.

page 4

The entirety of cardiac muscle does not truly contract in unison. There is apex to base excitation and contraction.

Responses: We agree with you on the point. However, when people say “the heart contracts in unison”, the term “unison” in such a context does not mean that the heart muscles of different anatomical regions really contract at the exact same moment, but simply means heart contraction during each single beat appears to be completed at the same moment.

“This property is gifted” is an awkward term. Please consider changing.

Responses: **“This property is gifted”** has now been changed to “This is because” (page 4, line 2)

“Individual cardiac muscle fibers” I would call them cells, not fibers. That terminology seems more appropriate for skeletal muscle or perhaps Purkinje fibers, but not working myocardium.

Responses: It is common the use the term cardiac fibers to describe cardiac cells simply because they are rod-shaped like fibers. Nonetheless, to address your concern, we have changed **“Individual cardiac muscle fibers”** to “Individual cardiac muscle cells” (page 4, line 3).

“come into the scene” is an awkward term.

Responses: We have changed **“come into the scene”** to “emerged” (page 4, line 17).

page 5

“manifested with” - perhaps replace with characterized by

Responses: We have changed **“manifested with”** to “with” by deleting “manifested”, as suggested (page 5, line 2).

“showed that the excitation conduction” - just simplify to showed that conduction velocity...

Responses: We have deleted **“excitation conduction”** (page 5, line 22).

page 7 and elsewhere

“cytoplasmic membrane” is a confusing term. What are you actually referring to?

Responses: It is well known that **“cytoplasmic membrane”** (or plasma membrane or cell membrane) is a double layer of lipids that surrounds the cytoplasm of a cell and separates the interior of a cell from its outside environment. We used this term to distinguish from other cell membrane structures (such as endoplasmic membrane, mitochondrial membrane, etc.).

page 8

“pronouncedly promoted” This needs to be changed - perhaps to significantly increased

Responses: We have changed **“pronouncedly promoted”** to **“significantly increased”**, as suggested (page 8, line 25). Additionally, we have also changed all other non-quantitative adjectives to **“significantly”**, where appropriate.

page 9

Its curious that there is a inverse relationship between CCR and Nav1.5. Please comment further on this observation.

Responses: We have commented on this observation as “These results indicated possible acceleration of intracellular conduction, presumably as a compensatory mechanism for the loss of intercellular conduction, and excluded Nav1.5 as a mechanistic link to the observed conduction disturbance in our model.” (page 9, lines 9-12). We do not add any more discussion on this issue because in the absence of experimental data, pure theoretical speculation will not add more meaningful information to our work, but will unnecessarily lengthen our manuscript.

Figure 2.

Please clarify how the optical mapping was performed. Was CV calculated during pacing or sinus rhythm? If pacing, where was the heart paced? It says conduction from septal apex- base, however the map seems to show activation from base toward apex?? CVs are not usually calculated as described in your methods section. In addition, perhaps using a color LUT without a large grey section in the middle would improve the images.

Responses: The procedures for our optical mapping studies are described under the subsection **“Determination of cardiac conduction velocity by optical mapping”** in METHODS. Specifically, (1) Our mapping study was conducted during pacing with an pacing electrode fixed at apex; hence, CV was calculated during pacing; (2) Our mapping images indeed well demonstrate that conduction was along the septal apex-base axis in control hearts as well as in our HF model; (3) We described two different methods of CV determination and both of them are standard procedures that have been commonly used for such a purpose. The CV values provided in our study were automatically calculated by the software used for the mapping measurements; (4) The color gradient is set by the software for the best discrimination of different colors thereby CVs.

In panel B, the sham control has an incidence of 2/13 with VT. In panel D, the sham control has an incidence of 7/20. This is quite a bit of variation for what should be two comparable control groups. Please explain.

Responses: Please be noted that the two “Shams” are under different conditions. The “Sham” in panel B is relative to surgical procedures to create heart failure; that is, the sham

animals underwent incision at the level of the suprasternal notch to allow direct access to the transverse aorta without opening chest. The “Sham” in panel D is relative to direct intramuscular injection of lentiviral constructs and the animals received open-chest surgery. Hence, the two “Shams” are not comparable. (The information is provided in METHODS of the main text and in Supplementary Materials)

Reviewer #2 (Remarks to the Author):

The authors' detailed response and additional studies are appreciated. No further concerns.

Responses: We highly appreciate your invaluable review of our manuscript that has helped us to improve our work and your satisfaction with our revision.

Reviewer #3 (Remarks to the Author):

Towards the original submission, all reviewers agreed that the findings regarding the functional impact of CCRR on cardiac conduction defect, arrhythmogenesis and overall pathological manifestation are interesting, novel and potentially important. The reviewers have also all identified similar issues regarding the mechanistic aspects of this report, namely, the validity of CCRR mediated regulation of Cx43 degradation via CIP85 interaction. In this revision, authors provided additional evidence and clarifications to further enforce the notion that CCRR is an important lncRNA to maintain cardiac conduction and contractile function. However, the revision did not provide adequate new data to address some of the previous concerns towards the mechanistic studies and their interpretations.

Responses: We thank you for your very positive comments on our work and helpful suggestions for improving our manuscript. We have conducted additional experiments, according to your suggestions, to address your concerns, as detailed below in a point-by-point manner.

1). The central hypothesis of this report (regarding mechanism) is that CCRR binds to CIP85, and blocks its negative regulation of Cx43 through endocytosis. However, other than weak binding between CCRR and CIP85, and very modest effect of CCRR expression on CIP85 distribution (cytosol vs. membrane), there are no definitive molecular evidence to support this molecular model. As authors agreed, this may be one of many possible mechanisms, but the text and conclusion leave the impression that the current model is the answer. Authors should either provide clear evidence that CCRR affects Cx43 degradation, and this effect is mediated through CIP85 and requires CIP85/CCRR interaction, or significantly revise the interpretation and conclusion.

Responses: We have to argue that biological processes or cellular functions may not be affected by transient, modest alterations of certain factors; nevertheless in reality they can well be significantly influenced by persistent, moderate alterations. This principle tells us that cellular function could be greatly affected by accumulation of small changes of certain factors (despite that people, even scientists are trained to see impressively large changes to expect the corresponding outcomes). However, we do entirely agree with you on that further insightful experiments could be done to make our data and conclusions more convincing. We must express our gratefulness to you for all your wonderful suggestions for improving our study. As explained below, we have followed your advices given in point 2 and performed the suggested experiments.

2). The following experiments are suggested examples to support the working hypothesis:

A. CCRR regulates CIP85 intracellular distribution through direct binding: Authors have identified potential interaction between the two via RNA-Protein Interaction Prediction (RPISeq) database and validated by experiments. However, there was no data to support the functional importance of such interaction. Authors should identify binding motifs on CCRR for CIP85 interaction and demonstrate the functional role of this motif in Cx43 regulation.

Responses: To address your concern, we have performed additional experiments as you have suggested. Specifically, we created a construct containing a short stretch of the putative binding motif on CCRR (that we named FD for Functional Domain). As anticipated, FD

reproduced the effects of the full-length CCRR in both animal and cellular studies, whereas the negative control and mutant FD failed to elicit such actions. It should be noted that the whole animal experiments on FD were performed in a mouse model of acute myocardial infarction (AMI) to meet the timeline for manuscript resubmission because HF model requires at least ten months to complete. Consistent with the results observed in our HF model, CCRR level was also significantly downregulated in AMI and such a downregulation also significantly enhanced the arrhythmogenic potential by slowing cardiac conduction and caused disruption of gap junction (see below for your information). More importantly, application of the lentiviral vector carrying FD (Lv-FD) rescued the reduced Cx43 expression, recovered the slowed conduction and reduced arrhythmias, in agreement with the effects of CCRR. In cellular studies, FD reproduced the effect of the full-length CCRR: increasing the diffusion of Lucifer yellow among AC16 cells (adult human ventricular cardiomyocytes). Moreover, similar to CCRR, FD upregulated the expression of Cx43. These new data are added to the text (**page 10, para 2 & page 11, para 1**), and **Figure 5a** and **Supplementary Figures 11 & 12**. Because the animal studies for FD were performed in MI model, we decided to place the data in Supplementary materials, instead of in the main text. It is worthy of mentioning that though the pathological models used for CCRR and FD are different, the findings would indicate that CCRR downregulation and its regulation of cardiac conduction may be a common mechanism for cardiac disease of different types.

In addition, we have also conducted RIP-Pulldown experiments to demonstrate the ability of FD to interact directly with CIP85. These new data have also been provided to the text (**page 10, lines 5-15**) and **Figure 4c & 4d**.

B. CCRR manipulation only yielded very modest changes in CIP85 expression. If authors argue about the importance of such modest changes in Cx43 expression, they should demonstrate if CCRR changes the interaction between Cx43 and CIP85. The prediction is CCRR blocks CIP85 interaction with Cx43 at functional level more so than at its expression.

Responses: We have conducted additional Co-IP experiments as you have suggested. The results are indeed in line with our original view that changes of CIP85 expression induced **by FD** caused significant decreases in the interaction between Cx43 and CIP85. The new data are incorporated into our revised manuscript (**page 11, lines 10-15**) and **Figure 5b**.

C. CIP85 mediated Cx43 expression, distribution and function: the current data provided only correlative relationship without clear indication on the directionality of the effect (forward insertion or degradation) of Cx43. This can be tested at least in cultured cells where the dependence of Cx43 downregulation on lysosomal activity can be directly demonstrated under CCRR manipulation.

Responses: We have followed your advice and carried out additional experiments on the dependence of Cx43 downregulation on lysosomal activity using co-immunocytostaining of Cx43 and lysosome marker Lamp1. The results presented in **Figure 5c** demonstrate that either CCRR or FD wiped out the overlapping staining of Cx43 and Lamp1 indicating the decreased recruitment of Cx43 into lysosomes for degradation, and simultaneously enhanced the Cx43 staining to the cytoplasmic membrane indicating the increased presence of Cx43 in the plasma membrane. The findings are described in page 11, lines 16-25 and page 12, lines 1-3.

3). There are still numerous places where non-quantitative adjectives are used such as “enormous”, “pronouncedly” to describe the quantitative changes.

Responses: We have now replaced the non-quantitative adjectives by “significantly”, except for the cases where only qualitative terms can be applied to the descriptions.

REVIEWERS' COMMENTS:

Reviewer #1 (Remarks to the Author):

The authors have addressed my concerns.

Reviewer #3 (Remarks to the Author):

No more comments as the revision significantly improved the validity of the conclusion.